# Carbon balance of a Finnish bog: temporal variability and limiting factors based on 6 years of eddy-covariance data

Pavel Alekseychik[1,2], Aino Korrensalo[3], Ivan Mammarella[2], Samuli Launiainen[1], Eeva-Stiina Tuittila[3], Ilkka Korpela[5] and Timo Vesala[2,4,6]

[1] Bioeconomy and Environment, Natural Resources Institute Finland, 00791 Helsinki, Finland

[2] Institute for Atmospheric and Earth System Research/Physics, Faculty of Science, P.O. Box 68, FI-00014 University of Helsinki, Finland

[3] School of Forest Sciences, University of Eastern Finland, P.O. Box 111, FIN-80101 Joensuu, Finland

[4] Institute for Atmospheric and Earth System Research/Forest Sciences, Faculty of Agriculture and Forestry, P.O. Box 27, FI-00014, University of Helsinki, Finland

[5] Department of Forest Sciences, University of Helsinki, P.O. Box 27, 00014 University of Helsinki

[6] Yugra State University, 628012, Khanty-Mansiysk, Russia

*Correspondence to:* Pavel Alekseychik (pavel.alekseychik@luke.fi)

## Abstract

Pristine boreal mires are known as substantial sinks of carbon dioxide ($CO_2$) and net emitters of methane ($CH_4$). Bogs constitute a major fraction of pristine boreal mires. However, the bog $CO_2$ and $CH_4$ balances are poorly known, having been largely estimated based on discrete and short term measurements by manual chambers, and seldom using the eddy-covariance (EC) technique.

Eddy-covariance (EC) measurements of $CO_2$ and $CH_4$ exchange were conducted in the Siikaneva mire complex in southern Finland in 2011–2016. The site is a patterned bog having a moss/sedge/shrub vegetation typical of Eurasian southern Taiga, with several ponds near the EC tower. The study presents a complete series of $CO_2$ and $CH_4$ EC flux ($F_{CH_4}$) measurements and identifies the environmental factors controlling the ecosystem-atmosphere $CO_2$ and $CH_4$ exchange. A 6-year average growing season (May-September) cumulative $CO_2$ exchange of $-61\pm24$ g C m$^{-2}$ was observed, which partitions into mean total respiration (Re) of $167\pm33$ (interannual range 146–197) g C m$^{-2}$ and mean gross primary production (GPP) of $228\pm46$ (interannual range 193–257) g C m$^{-2}$, while the corresponding $F_{CH_4}$ amounts to $7.1\pm0.7$ (interannual range 6.4–8.4) g C m$^{-2}$. The contribution of October-December $CO_2$ and $CH_4$ fluxes to the cumulative sums was not negligible based on the measurements during one winter.

GPP, Re and $F_{CH_4}$ increased with temperature. GPP and $F_{CH_4}$ did not show any significant decline even after a substantial water table drawdown in 2011. Instead, GPP, Re and $F_{CH_4}$ were limited in a cool, cloudy and wet growing season of 2012. May-September cumulative net ecosystem exchange (NEE) of 2013–2016 averaged at about $-73$ g C m$^{-2}$, in contrast with the hot and dry year 2011 and the wet and cool year 2012. Suboptimal weather likely reduced the net sink by about 25 g C m$^{-2}$ in 2011 due to elevated Re, and by about 40 g C m$^{-2}$ in 2012 due to limited GPP. The cumulative growing season sums of GPP and $CH_4$ emission showed a strong positive relationship.

The EC source area was found to be comprised of eight distinct surface types. However, footprint analyses revealed that contributions of different surface types varied only within 10–20% with respect to wind direction and stability conditions. Consequently, no clear link between $CO_2$ and $CH_4$ fluxes and the EC footprint composition was found despite the apparent variation of fluxes with wind direction.

**1. Introduction**

Natural mires are an important element of the Boreal and Arctic carbon cycle due to the vast amounts of carbon stored in peat and their sensitivity to environmental changes (Gorham 1991, Charman et al. 2013). Over the last 10–14 thousand years, the northern mires have provided substantial climatic cooling (Frolking and Roulet 2007). It is expected, however,

that the rising air temperature will likely drive an increase in the emission of methane ($CH_4$) (Zhang et al. 2017) and carbon dioxide ($CO_2$) (Laine et al. 2019). As the effects of rising temperature are strongly dependent on water table level (Davidson and Janssens 2006, Buttler et al. 2015, Laine et al. 2019), the precipitation extremes in northern latitudes (IPCC 2013) will contribute to the variability and uncertainty of any predictions.

Pristine bogs, i.e. primarily rain-fed, oligotrophic peatlands with a developed microtopography and often populated by

trees, constitute a major and diverse class of boreal mires (Seppä 2002). The response of their GHG balances to the environment is therefore of utmost interest. The complex surface patterning of a bog and the need for continuous measurements favours the application of area-integrating eddy-covariance (EC) measurements (Baldocchi 2008), a technique providing an estimate of vertical net flux of scalars and heat over a large (~1–100 ha) source/sink area (flux footprint) that typically envelops all representative microsites. However, most boreal bog GHG balance estimates to date

have been produced using flux chambers (Bubier et al. 1993, Alm et al. 1999, Waddington and Roulet 2000, Bubier et al. 2003, Laine et al. 2006, Saarnio et al. 2007, Korrensalo et al. 2019), which requires ecological understanding of the spatial variability of a strongly patterned ecosystem for reliable upscaling (Laine et. 2009, Riutta et al. 2007) which requires very labour-intensive and lengthy field campaigns. Therefore, it would be of high interest to examine a multiannual EC record from a bog site.

The strong ecological diversity of bogs is reflected in the previous bog eddy-covariance studies, which examined a temperate wooded bog (Fäjemyr; Lund et al. 2007, Lund et al. 2012), temperate shrub bog (Mer Bleu; Lafleur et al. 2001, Lafleur et al. 2005, Roulet et al. 2007, Brown 2014), boreal treed/low shrub bog (Attawapiskat river; Humphreys et al. 2014), boreal shrub bog (Kinoje Lake; Humphreys et al. 2014), boreal raised bog (Tchebakova et al. 2015), boreal collapse scar bog (Bonanza Creek Experimental Forest; Euskirchen et al. 2014), boreal raised patterned bog (Mukhrino;

Alekseychik et al. 2017), temperate patterned *Sphagnum*/shrub bog (Arneth et al. 2002), and boreal open patterned bog (Arneth et al. 2002). Due to such a broad range of vegetation and climate, identifying the "typical" bog GHG balance and its environmental controls presents a certain challenge.

The previous EC and chamber estimates indicate that boreal bogs typically demonstrate a small to moderate annual sink of $CO_2$ and a relatively small source of $CH_4$, the flux rates being similar to those observed in fens. Net loss of carbon has

been observed in exceptionally dry summers (Alm et al. 1999, Waddington and Roulet 2000). The annual (or growing season) net $CO_2$ exchange in bogs across the entire boreal region varies typically from +30 to -100 g C m$^{-2}$ a$^{-1}$ (e.g. Alm et al. 1999, Waddington and Roulet 2000, Bubier et al. 2003, Laine et al. 2006, Saarnio et al. 2007, Tchebakova et al. 2015, Lund et al. 2010, Roulet et al. 2007, Lafleur et al. 2003, Korrensalo et al. 2019,), which splits into gross primary productivity (GPP) and ecosystem respiration (Re) both typically in the range of 200–500 g C m$^{-2}$ a$^{-1}$. The fairly wide

spread in these numbers is attributed to the variation in site-specific (vegetation, hydrology, peat structure; see Moore at al. 2002 and Korrensalo et. al. 2018 for dry and wet bog, respectively) and external (climate, weather; Moore et al. 2002 and Laine et al. 2006 for continental and maritime big, respectively) factors. Studies extending over several years reveal that bog $CO_2$ balance is sensitive to temperature and water table level (WTD) (Lafleur et al. 2003, Rinne et al. 2020). Hot and dry weather suppresses bog photosynthesis and promotes respiration (Alm et al., 1999; Lund et al., 2012, Euskirchen et al. 2014, Arneth et al. 2002, Tchebakova et al. 2015, Lund et al. 2012). Under favourable conditions, which seem to consist of warm temperatures, ample sunshine, sufficient moisture but no long-term WTD rise, bogs can show a high growing season net $CO_2$ uptake of 100–200 g C m$^{-2}$ (e.g. Friborg et al. 2003, Alekseychik et al. 2017).

In wetlands, the total methane emission is a sum of diffusion through soil matrix, ebullition and plant transport, each associated with a set of environmental controls (Lai 2009, Dorodnikov 2011, Ström et al. 2015, Korrensalo 2018, Männistö et al. 2019, Riutta et al. 2020). Peat temperature is known to be the primary driver of $CH_4$ production (e.g. Dunfield et al. 1993). About 50–90% of the produced methane is oxidized in the oxic zone before it can reach the atmosphere (King et al. 1990, Fenchner and Hemond 1992, Whalen and Reeburgh 2000), so WTD is *a priori* an important driver. While wet surfaces were considered to be the highest emitters (Bubier et al. 2005), recent work has shown the maximum fluxes to occur at intermediate WTD microsites (Turetsky et al. 2014, Rinne et al. 2018). There contradictions call for multiyear studies from patterned bogs to reveal the temporal controls of methane emissions.

At present, the lack of multi-year eddy-covariance studies in boreal bogs does not allow to rank the environmental controls by their importance for $CH_4$ emission. It was previously observed that northern bogs typically emit from 0 to 20 g C m$^{-2}$ annually in the form of methane (Vompersky et al. 2000, Friborg et al. 2003, Roulet et al. 2007). The bogs with developed ridge-hollow microtopography were identified as the mire types with the highest $CH_4$ emission spatial variability in western Siberia (Kalyuzhny et al. 2009); a high spatial heterogeneity was also found in a Canadian bog study (Moore et al. 2011).

The available shorter datasets from bogs, and the more abundant data from fens, do shed some light on the possible controls. Methane net efflux ($F_{CH4}$) is strongly correlated with GPP; to which extent this is due to accelerated rhizospheric $CH_4$ production as a result of photosynthate input or enhancement of $CH_4$ transport by vascular plants, is not clear (Bellisario et al. 1999, Rinne et al. 2018). The WTD control is equally unclear, with reports of both an optimum WTD (e.g. Rinne et al. 2018) and a limitation in $CH_4$ efflux at WTD drawdown (Glagolev et al. 2001, Kalyuzhny et al. 2009). Euskirchen et al. (2014) show that only a drought or considerable strength is able to limit bog $CH_4$ emission. $CH_4$ ebullition is prompted by drops in atmospheric pressure but is reduced by drops in peat temperature (Fechner-Levy & Hemond 1996).

The EC technique is usually assumed to provide flux estimates representative of the studied ecosystem (e.g. Aubinet et al. 2012). However, in heterogeneous sites this may not be the case. The bog surface cover heterogeneity has several characteristic spatial scales, including vegetation community (1 m$^2$), microsites, e.g. hummocks and hollows (1–10 m$^2$), and larger formations, such as ponds and ridges (50–1000 m$^2$). These surface elements have been shown to have significantly different surface-atmosphere GHG exchange rates (Alm et al. 1999, Repo et al. 2007, Maksuytov et al. 2010, Kazantsev and Glagolev 2010). In order to understand the output of an EC system in such a heterogeneous site, one needs to estimate the contributions of the surface cover types to the total flux. Due to the specifics of atmospheric transport, the EC flux is strongly influenced by sources close to the EC tower (Vesala et al. 2008), but also by the larger scale general surface composition in different wind direction sectors. The possible effect of the resulting footprint variation on the EC data calls for detailed inquiry (e.g. Tuovinen et al. 2019).

In the present study, we report six years of $CO_2$ and $CH_4$ fluxes measured by EC at a raised patterned bog area of the Siikaneva mire, Southern Finland. The long-term data is used to analyze the responses of bog-atmosphere $CO_2$ and $CH_4$ exchange to environmental forcing during growing season (May-September), and to provide growing season balances. Specifically, we aim to:

1) Quantify $CO_2$ and $CH_4$ balances of a boreal bog ecosystem on seasonal and interannual timescales;

     2) Identify the environmental controls responsible for seasonal and interannual variability of bog carbon exchange;

     3) Inquire whether signals from surface heterogeneity within the footprint can be associated with the observed $CO_2$ and $CH_4$ fluxes.

**2. Materials and methods**

**2.1 Site description and previous research**

     The Siikaneva-2 site (61.8° N, 24.2° E) is situated in a patterned ombrotrophic bog within the larger Siikaneva mire

complex in Southern Finland (Fig. 1). The microtopography is dominated by rows of hummock strings (Fig. 1a,b,d). The areas with lower elevation are considerably varied, being composed of hollows and lawns, moss-free mud bottoms and ponds. The dominating moss species are *Sphagnum rubellum*, *S. papillosum*, *S. fuscum*, while the most widespread vascular plants are *Rhynchospora alba*, *Andromeda polifolia*, *Calluna vulgaris* and *Rubus chamaemorus*; a few small Scots pines (*Pinus sylvestris*) grow on the ridges.

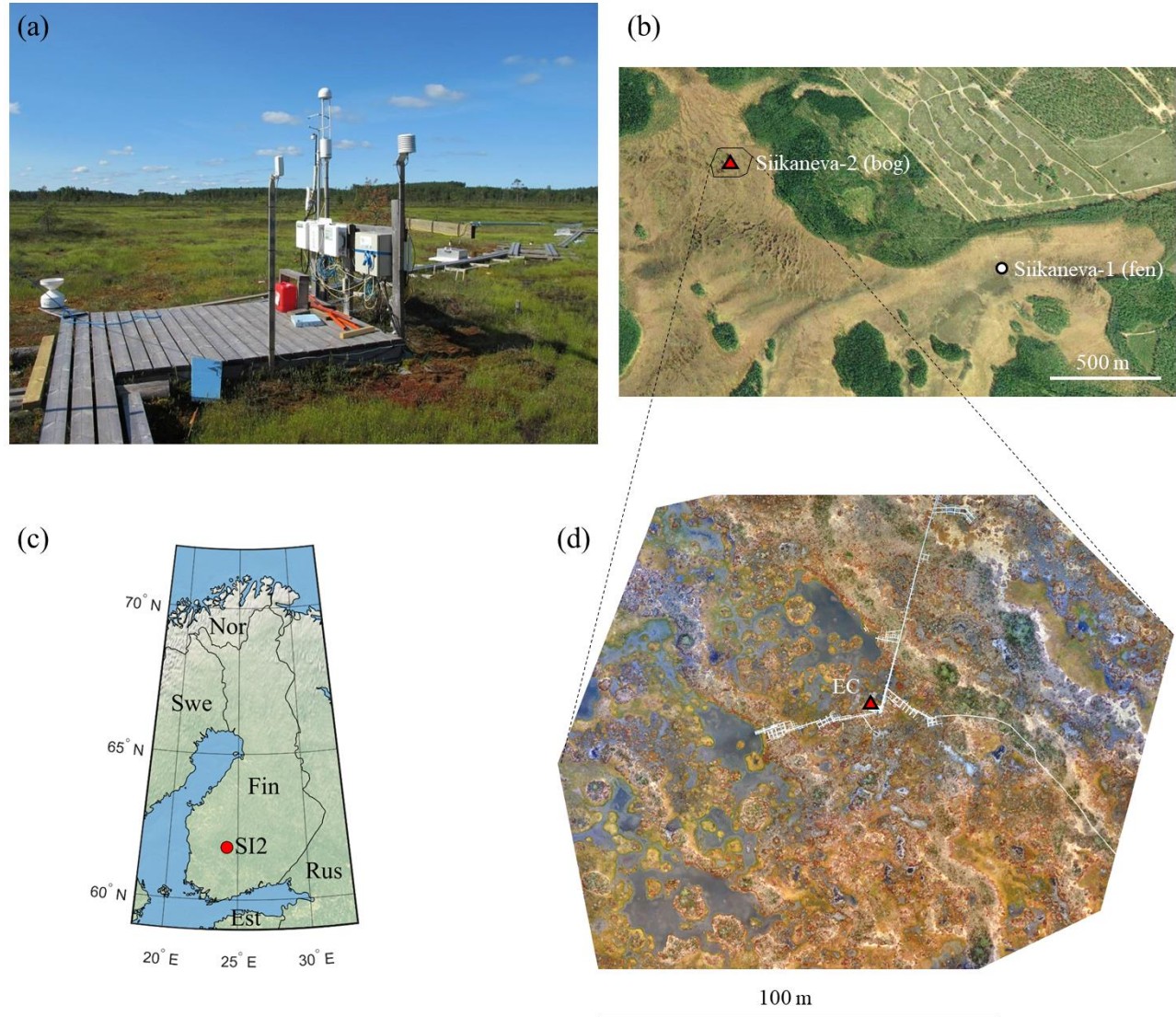

**Figure 1:** (a) Photo of the eddy-covariance tower and meteorological setup facing south-west. (b) Map of the Siikaneva mire with the bog (SI2, this study) and fen (SI1, Rinne et al. 2018) sites marked. (c) Map of Finland showing the location of the Siikaneva-2 site. (d) Aerial RGB orthomosaic showing the main EC footprint contribution zone, based on the imagery obtained on 18 June 2018 during a dry period.

The pond depths range from 0 to 2 m. The temporal variations in the bog water level lead to the variation in pond size and occasionally result in inundation of the mud bottoms. The 4-year average (2012-2015) peak total vascular plant leaf area index (LAI) was 0.35 $m^2$ $m^{-2}$ and the peak aerenchymous LAI was 0.24 $m^2$ $m^{-2}$. Details on microform types and vegetation composition is given in Korrensalo et al. (2016) and Korrensalo et al. (2017).

The site is highly heterogeneous due to the patchy vegetation cover. The western sector of the EC footprint is dominated by ponds, hollows and lawns, while in the eastern sector one encounters several well-defined ridges (Fig. 1d). The nearest forest edge lies in 150 m NE of the EC tower.

### 2.2 Measurements

#### 2.2.1 Eddy-covariance measurements

EC measurements of $CO_2$, $CH_4$, sensible and latent heat fluxes were conducted at the site in the years 2011–2016 using a METEK USA-1 anemometer, a LiCor LI-7700 open-path $CH_4$ analyzer, and a LiCor LI-7200 enclosed path $CO_2$ & $H2O$ analyzer mounted at 2.4 m height above the moss surface. The EC raw data were processed using EddyUH software (Mammarella et al., 2016) following standard schemes and quality control protocols (Nemitz et al. 2018, Sabbatini et al. 2018). The $CH_4$ flux data at Relative Signal Strength (RSSI) < 20 were excluded from analysis based on the regression of $F_{CH4}$ versus RSSI. A friction velocity ($u_*$) filter of 0.1 m $s^{-1}$ was applied based on the observed behavior of the normalized $CH_4$ and $CO_2$ fluxes under low turbulence conditions (Appendix A). See Table 1 for the proportion of the 30-min average $CO_2$ and $CH_4$ fluxes remaining after quality control and $u_*$-filtering. We conventionally define May-Sep as the growing season and Oct-Apr as non-growing season.

**Table 1:** Fraction of 30-min EC fluxes of $CO_2$ and $CH_4$ remaining after the quality checks and $u_*$-filtering, in % of the specified period.

|         | May-September | | June-August | |
|---------|---------|---------|---------|---------|
|         | $F_{CO2}$ | $F_{CH4}$ | $F_{CO2}$ | $F_{CH4}$ |
| 2011    | 27 | 21 | 41 | 33 |
| 2012    | 63 | 44 | 55 | 33 |
| 2013    | 18 | 37 | 24 | 34 |
| 2014    | 52 | 43 | 72 | 57 |
| 2015    | 45 | 40 | 59 | 51 |
| 2016    | 53 | 31 | 75 | 45 |
| average | 43 | 36 | 54 | 43 |

### 2.2.2 Auxiliary measurements

Meteorological and environmental measurements were conducted next to the EC tower. Peat temperature ($T_p$) profile was measured by Campbell 107 Thermistor sensors at 5, 20, 35 and 50 cm depths since July 2011. In April 2012, the auxiliary measurements were expanded with a net radiation sensor Kipp & Zonen NR Lite2, air temperature ($T_a$) and relative humidity (RH) sensor Campbell CS215, WTD sensor Campbell CS451 and tipping bucket rain gauge ARG-100. The peat temperature profile and the water table sensor were installed in a lawn microform near the EC tower.

The peat temperature at 20 cm depth ($T_p20$) and WTD were gap-filled using regressions with the data from the Siikaneva-1 fen station (SI1, Rinne et al. 2007, Rinne et al. 2018) 1.2 km SE from the site. Measurements from the SMEAR-II station (Hyytiälä, 7 km away) were used to gap-fill $T_a$ and RH and supply the complete timeseries of photosynthetic active radiation (PAR). Precipitation rate time series was constructed as a combination of observations at SMEAR-II site and Finnish Meteorological Institute weather station in Hyytiälä.

Annual campaigns for leaf area index (LAI) measurement were undertaken at the site in 2012–2015. The representative microforms were covered by three replicate plots 60 x 60 cm in size, 18 in total. Within each plot, five small subplots were defined for the manual measurement of leaf number. This measurement was made approximately twice a month throughout the growing season, and simultaneously, average leaf size of each species was defined with a scanner. To

obtain the leaf area of the subplots, the leaf number was multiplied by the average leaf area. These community-specific LAI were then averaged and weighted by their area fractions within EC footprint to yield the ecosystem-scale value (Korrensalo et al. (2017). The years of 2011 and 2016, when LAI was not measured, were filled with the 2012–2015 average LAI, as no LAI anomalies were visually detected in either 2011 or 2016 (Aino Korrensalo, personal communication, 2021).

### 2.2.3. Aerial imaging

Airborne survey combining Lidar scanning and aerial photography was conducted by helicopter in May 2013. The data at the original resolution of 4 cm/pixel were processed and converted into a surface cover type map and a digital elevation map (Fig. 2). As the individual microforms measure in area roughly 1–2 m$^2$, and in order to mitigate the motion blur present in the images, the resolution of all maps was coarsened to 1 m$^2$ pixel$^{-1}$.

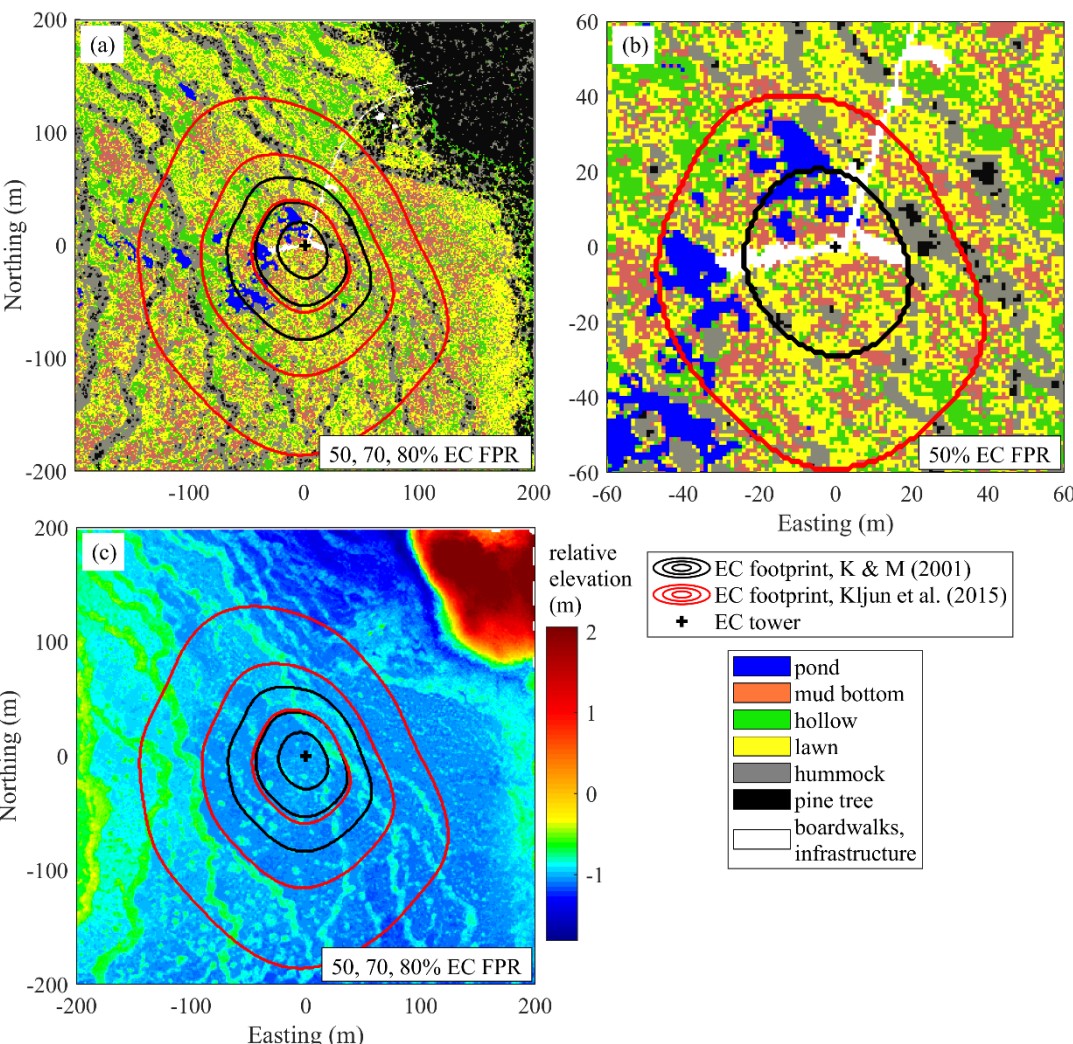

**Figure 2:** Site characteristics obtained from the helicopter survey in May 2013. a) Surface cover map at 1 m resolution showing a 400 x 400 m patch of landscape centered on the EC tower. The cumulative Kormann & Meixner (2001) and Kljun et al. (2015) flux footprints are given by 50, 70 and 80% contribution contours. b) Close-up map, focusing on the

205 50% cumulative footprint zones. c) Digital elevation model derived from Lidar scan. Note that the 70% Kljun et al. (2015) isoline coincides with the 50% Kormann & Meixner (2001) isoline. FPR – footprint.

The proportion of each microform for the whole extent of the map (400 x 400 m, centered on the EC tower) and their proportions weighted by the two footprint models were roughly similar (Table 2). Note that due to the proximity of the
210 EC tower to the ponds (as was intended on the site planning stage) and boardwalks, their share in the EC flux source are comparatively higher than the 400 x 400 m map means. The boardwalks extending in the western and eastern directions from the EC tower raft were built in April–June 2012, after the first measurement season. These building works caused only insignificant ecosystem damage, as no changes in the peat surface and vegetation cover around the new boardwalks were apparent. Also note that when the boardwalks some 30–40 cm wide are rescaled to the resolution of 1 m, their area
becomes somewhat exaggerated.

**Table 2:** Proportions of the surface types and microforms, expressed as area percentage (first column) or weighted by the cumulative footprints, Kormann & Meixner (2000) and Kljun et al. (2015) (second and third columns).

| | Areal cover (%) within the entire map (200 x 200 m) | Contribution (%) for K&M cumulative footprint within 80% contribution zone | Contribution (%) for Kljun cumulative footprint within 80% contribution zone |
|---|---|---|---|
| pond | 2.3 | 7.7 | 9.1 |
| mud bottom | 15.3 | 19.7 | 24.2 |
| hollow | 18.8 | 14.4 | 10.3 |
| lawn | 27.3 | 27.3 | 26.3 |
| high lawn | 7.1 | 6.9 | 7.0 |
| hummock | 10.5 | 10.5 | 9.6 |
| high hummock | 18.3 | 8.2 | 5.5 |
| boardwalks | 0.4* | 5.3* | 8.0* |

*overestimate*

### 2.3 EC flux footprint modeling

EC scalar flux footprints were calculated using the popular Kormann & Meixner (2001) and Kljun et al. (2015) models. The footprint lengths are very dependent on roughness length $z_0$, necessitating its careful calculation and quality control.
The 30-min average values of $z_0$ were calculated using the expression

$$z_0 = z \exp\left(\frac{-\kappa U}{u_*} - \psi_m\left(\frac{z}{L}\right)\right), \tag{1}$$

where z is the height above ground, $\kappa$ the Von Kármán constant, U the wind speed at level z, $u_*$ the friction velocity at level z, $\psi_m(z/L)$ is the stability correction function defined, following Beljaars & Holtstag (1991), as

$$\psi_m = 2log\left(\frac{1+x}{2}\right) + log\left(\frac{1+x^2}{2}\right) - 2\arctan x + \frac{\pi}{2} \qquad \left(\frac{z}{L} < 0\right) \qquad (2)$$

$$\psi_m = -a\frac{z}{L} - b\left(\frac{z}{L} - \frac{c}{d}\right)exp\left(-d\frac{z}{L}\right) - \frac{bc}{d} \qquad \left(\frac{z}{L} > 0\right) \qquad (3)$$

with $x = \left[1 - \left(16\frac{z}{L}\right)\right]^{1/4}$, a = 0.7, b = 0.75, c = 5 and d = 0.35. 30-min $z_0$ values were used to model the footprint for each 30-min period so that to account for the variation in $z_0$ due to the temporal changes in stability (Zilitinkevich et al. 2008) and directional variation in surface roughness. It was necessary to filter the calculated 30 min $z_0$ values for very stable nocturnal conditions, which initiate a decoupling phenomenon. At nighttime thermal decoupling, which in Siikaneva typically happens below the EC measurement height (unpublished data), spikes in $U/u_*$ are observed which can be interpreted as airflow losing contact with the surface; therefore, the $z_0$ values at $U/u_* > 12$ and/or $u_* < 0.1$ m s$^{-1}$ were eliminated. High 30-min average $z_0$ values may also occur at strongly convective conditions, when high instability is combined with low wind speeds, but those values are retained, with only the extreme values $z_0 > 3$ m being excluded. The $z_0$ remaining after such filtering were from 0 to 3 m, with 80% of the values being between 0 and 0.1 m and the most probable value being 0.03 m. The displacement height was set to zero as the surface roughness elements are small and sparse.

Footprint estimates were calculated on a 1 x 1 m grid in order to match the resolution of the surface cover map. The footprint nodes lying further than 200 m away from the origin (coinciding with the location of the EC tower) were excluded from analyses. Such an exclusion of the map corners was necessary since, should they have been preserved, there would have been a bias between the footprints extending towards the corners and those extending in N, E, S and W directions due to the difference in the number of nodes. Finally, since a fraction of the EC signal (roughly 10–20%) comes from beyond a 200 m distance from the tower, all footprint values within this domain were normalized by their cumulative sums in order to set them to unity.

### 2.4 Modeling the $CO_2$, $CH_4$ fluxes and EC flux gap-filling

The EC fluxes were modeled and gap-filled using the method similar to that in Alekseychik et al. (2017). In the initial modeling trials, Re, $F_{CH4}$ and GPP showed clear responses to peat temperature and PAR but much more complex relationships with WTD, making it impossible to capture the combined effect of environmental drivers on fluxes by fitting a single function to all data. As the dataset includes both long (>15 days, mainly on the edges of the growing season) and short (<=15 days) gaps, a special modeling approach was needed to fill the long gaps with robust, defensible estimates of GHG exchange, and closely imitate the fluxes during shorter gaps. For gap distribution, see Fig. 5.

The models to fill the long gaps were obtained by fitting the standard $Q_{10}$ and Micaelis-Menthen-type expressions to all available quality-controlled Re, GPP and $F_{CH4}$ data,

$$Re_{mod} = Re_{ref} \, Q_{10\,CO2}^{\left(\frac{T_{p5}-12}{10}\right)} \qquad (4)$$

$$F_{CH4mod} = F_{ref} Q_{10\,CH4}^{\left(\frac{T_{p20}-12}{10}\right)}$$ (5)

$$GPP_{mod} = \frac{P_{max} PAR}{k+PAR}\left(aT_{p\,5cm} + b\right)$$ (6)

where $Re_{ref}$ and $F_{ref}$ are the reference ecosystem respiration and $CH_4$ flux (model value at 12 ˚C), $P_{max}$ the maximum photosynthesis, k the value of PAR at GPP = $0.5P_{max}$, PAR the photosynthetically active radiation, $Q_{10\,CO2}$ and $Q_{10\,CH4}$ the temperature sensitivity parameters for respiration and $CH_4$ flux, respectively. The resulting model parameter values are summarized in Table 3. The Micaelis-Menthen expression for GPP is expanded with a peat temperature module as it was

275 found to improve the fit, adding the linear function parameters $a$ and $b$. $T_{p20}$ was chosen as the driver of $F_{CH4}$ as it performed slightly better (about 5%) than $T_{p5}$ in terms of model $R^2$, RMSE and SSE. A reference temperature of 12˚C was used for both Re and $F_{CH4}$ (Eqs. 4 and 6) as a representative peat temperature at the site.

**Table 3:** Parameters of the fits for Re, GPP and $F_{CH4}$ made using all quality-controlled data. The 95% confidence intervals
are given in parentheses.

| | | |
|---|---|---|
| | **$R_{ref}$ (µmol CO$_2$ m$^{-2}$ s$^{-1}$)** | **$Q_{10\,CO2}$** |
| Re (Eq. 4) | 0.73 (0.68–0.78) | 3.39 (2.87–3.90) |
| | **$F_{ref}$ (µmol CH$_4$ m$^{-2}$ s$^{-1}$)** | **$Q_{10\,CH4}$** |
| $F_{CH4}$ (Eq. 5) | 0.038 (0.037–0.040) | 4.91 (4.43–5.39) |
| | **$P_{max}$** | **k** |
| GPP (Eq. 6) | 4.40 (4.39–4.41) | 222 (199–245) |
| | **a** | **b** |
| | 0.074 (0.070–0.078) | -0.10 (-0.15– -0.06) |

To fill the short (<15 days) gaps, another set of models was constructed using the sliding time window approach. This approach allows for closely imitating the weather-dependent changes in fluxes without a detailed prior knowledge of the
285 drivers. For Re and $F_{CH4}$, Eqs. 4-5 were once again used; that for GPP, however, was simplified by eliminating the temperature module, as the information on the temperature variations would be implicit in the time-resolved model parameters:

$$GPP_{mod} = \frac{P_{max} PAR}{k+PAR}$$ (7)

The sliding time-window models were recalculated with a daily step, using the data from a period of 15 days for Re, 10 days for GPP, and 5 days for $F_{CH4}$. The resulting daily values of $Re_{ref}$, $F_{ref}$, $P_{max}$ and k were linearly interpolated to each 30 min period. Section 3.3 and Fig. 6 provide the calculated parameter time series. The time-resolved model parameters are a valuable by-product of this approach, as their variations on weekly to seasonal timescales can provide clues on flux

controls, similar to normalized fluxes used in some other studies. Finally, Re, GPP and $F_{CH4}$ are gap-filled using a combination of the two above models: sliding time window model for short gaps, and general-fit models for long gaps. The sliding window model performed well. The $R^2$ of the median daily model vs. measured fluxes was 0.71 for $F_{CH4}$ and 0.67 for NEE, which is similar to what Raivonen et al. (2017) obtained using a process-based model HIMMELI with the data from the nearby Siikaneva-1 fen. However, the model in this study seems to capture the mean level of fluxes better than Raivonen et al. (2017), the mean ratio of the model/measured flux being 0.99 for $F_{CH4}$, 0.99 for GPP, and 1.02 for Re. Thus, the above model is fully adequate for the purposes of this study − gap-filling and imitating the mean daily to seasonal course of surface GHG exchange.

## 3 Results

### 3.1 Environmental conditions

The climate is south boreal, with the mean 30-year average annual precipitation (liquid equivalent) of 711 mm and air temperature 3.5 ˚C (Table 4). The summer is typically warm and relatively dry, having a 30-year average air temperature of 14.6 ˚C and precipitation of 252 mm. The weather in 2011-2016 represented a range of conditions from warm/dry/sunny to cool/moist/cloudy (Fig. 3). Most of the years were warmer and drier than the 30-year average. Fig. 4 shows the seasonal variation in the main environmental drivers. The year 2012 was atypical due to its low temperatures and ample precipitation, whereas another cool year 2015 had a sunny but dry autumn and an early drop in LAI (Fig. 4g, h). Normalized LAI was rather similar in the rest of the study years. 2011 had an untypically dry spring and summer, which resulted in the lowest instantaneous WTD of -25 cm and the lowest mean summertime WTD of -20 cm. The dry conditions of 2011 were also evident in the sharply increased diurnal amplitude of the peat temperature at 5 cm depth, implying the top peat layer and moss capitula may have been desiccated. The highest average growing season WTD of -7 cm was observed in 2012 (Fig. 4e).

**Table 4:** Average annual and summer air temperature, precipitation and water table depth in comparison with 30-year averages.

| Period | Annual | | June-August | | |
|---|---|---|---|---|---|
| | $T_a$ (˚C) | precip. (mm) | $T_a$ (˚C) | precip. (mm) | WTD (cm) |
| 2011 | 5.2 | 777 | 16.6 | 261 | -20 |
| 2012 | 3.3 | 925 | 14.1 | 310 | -7 |
| 2013 | 5.1 | 632 | 15.9 | 240 | -15 |
| 2014 | 5.0 | 642 | 15.7 | 314 | -12 |
| 2015 | 5.6 | 678 | 14.3 | 230 | -11 |
| 2016 | 4.4 | 784 | 14.8 | 356 | -10 |
| long-term | 3.5 [a] | 711 [a] | 14.6 [a] | 252 [a] | -12 [b] |

a 1981-2010, Pirinen et al. (2012)

b The average of 2011–2016, as no longer-term data exist.

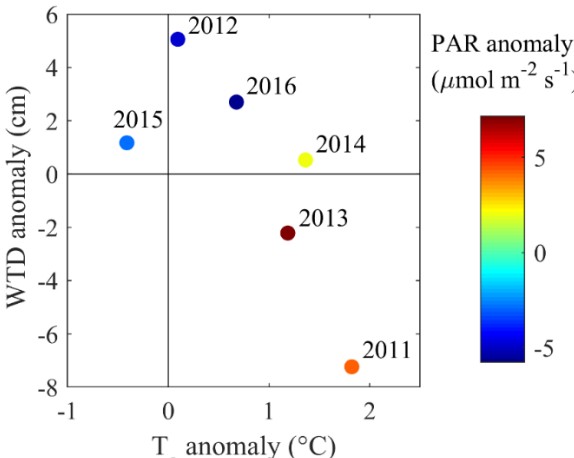

**Figure 3:** June-August deviations in water table depth, air temperature and PAR, calculated as difference from the 6-year averages for WTD and PAR and 30-year averages for $T_a$.

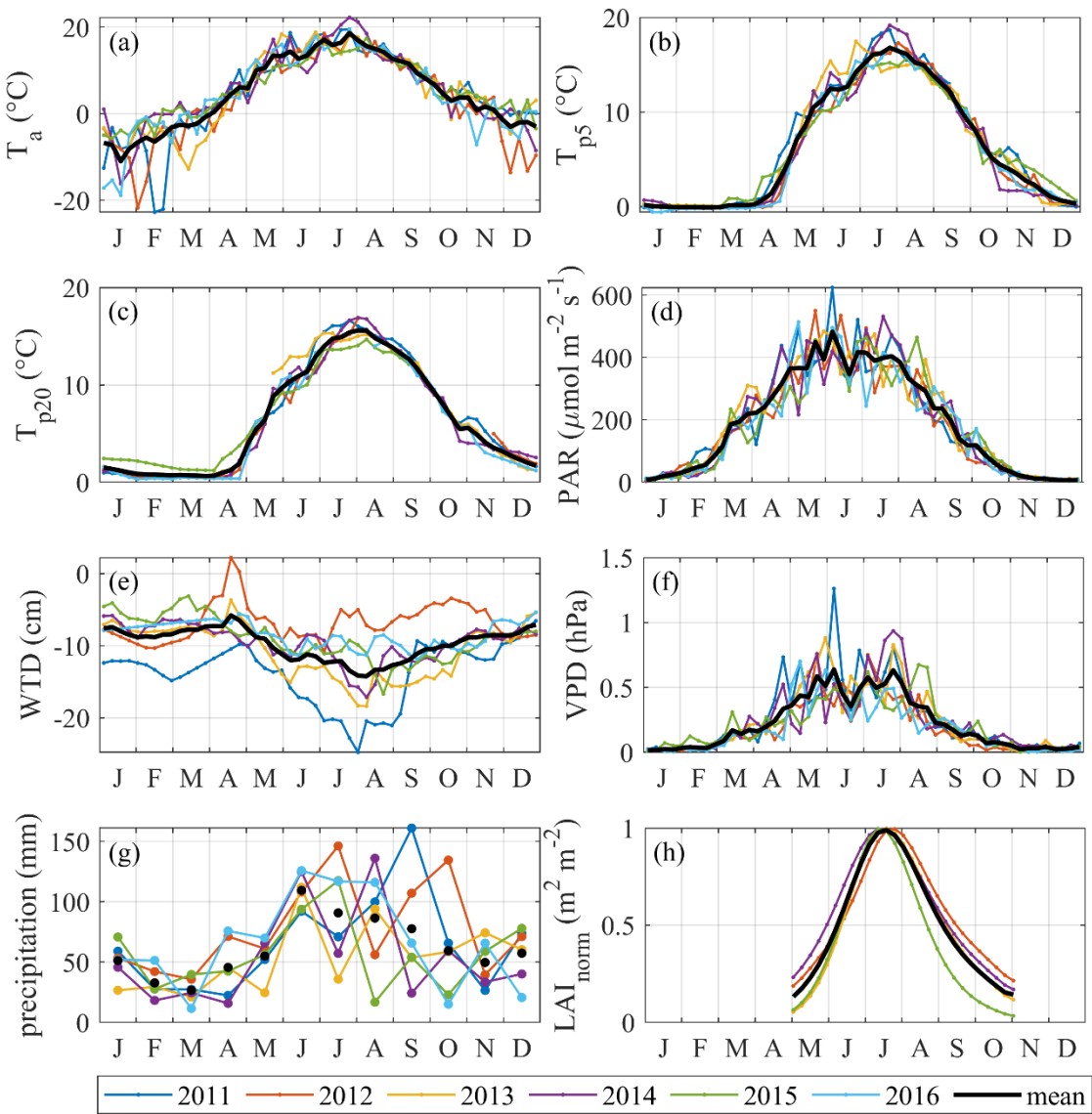

**Figure 4:** Time-series of potential drivers of NEE, GPP, Re and $F_{CH4}$. LAI is normalized by the annual peak value. The color lines in (a), (b), (c), (d), (f) represent weekly means, whereas in (e, h) they give daily average WTD and in (g) monthly precipitation sums. The black lines in (a–f) show the mean annual course of all years; the black markers in (g) are the monthly average precipitation of all years.

## 3.2 Seasonal variability in the CO₂ and CH₄ fluxes

The EC flux time series shown in Fig. 5 reveal the typical seasonality with pronounced summer maxima. The measurements were mostly conducted during the growing season, with some autumn and winter periods covered in 2011–2012 and 2014–2015. Both the $CO_2$ and $CH_4$ fluxes are at their highest during the growing season from May to September. While the seasonal curve of NEE does not appear to show a marked deviation from the typical domed shape, its daily variation responded strongly to environmental conditions on a weekly to biweekly scale, with the beneficial conditions resulting in peak daily mean NEE of about -1…-1.5 $\mu mol\ m^{-2}\ s^{-1}$, and cool/cloudy/rainy weather reducing that to zero. The seasonality of the partitioned $CO_2$ fluxes varied markedly between the years. 2011 and 2014 yielded higher seasonal

peaks in GPP and Re than in the other years, corresponding with the periods of warmest weather. Maximum GPP reached 4.3–4.5 µmol m$^{-2}$ s$^{-1}$ in 2011 and 2014, which is substantially higher than 3.0–3.1 µmol m$^{-2}$ s$^{-1}$ observed in the other years. Correspondingly, the maximum daily mean respiration reached 3.5 µmol m$^{-2}$ s$^{-1}$ in 2011 and 2.7 µmol m$^{-2}$ s$^{-1}$ in 2014, while the other years showed 1.6–1.9 µmol m$^{-2}$ s$^{-1}$ at the peak. The summer peaks in CH$_4$ flux in 2011 and 2014 closely matched those in GPP and Re and also resulted in the highest daily averages reaching 0.1 µmol m$^{-2}$ s$^{-1}$. In the other years, CH$_4$ flux daily means reached 0.07–0.08 µmol m$^{-2}$ s$^{-1}$ with an overall smoother seasonal maximum.

The importance of the non-growing season fluxes (Oct–Apr) was also analyzed. The spring peak of CH$_4$ emission associated with the thaw period in April and May contributes only a minor, although non-negligible fraction of spring-summer total. In 2012, elevated net emission lasted for about 8 days (25 April–3 May), and in 2013, 10 days (22 April–5 May). These periods supplied roughly 4% of the cumulative CH$_4$ emission of 25 April – 31 August 2012 and 22 April – 31 August 2013. Based on partially covered winters of 2011-2012 and 2014-2015, the cumulative wintertime season fluxes were relatively small but non-negligible as well. In 2011, the Oct–Nov contributions to May-Nov cumulative fluxes were as follows: 12% for F$_{CH4}$, 4% for GPP, and 8% for Re. In 2015, the corresponding values were 14%, 5% and 13%. Finally, in 2014, Oct–Dec contributed 10% of F$_{CH4}$, 4% of GPP, and 13% of Re to the May–December totals (see a summary in Table 5).

**Table 5:** contributions of the non-growing season fluxes to the total measured period.

| flux | 2011 | 2014 | 2015 |
|---|---|---|---|
| | Oct–Nov/May–Nov (%) | Oct–Dec/May–Dec (%) | Oct–Nov/May–Nov (%) |
| F$_{CH4}$ | 12 | 10 | 14 |
| GPP | 4 | 4 | 5 |
| Re | 8 | 13 | 13 |

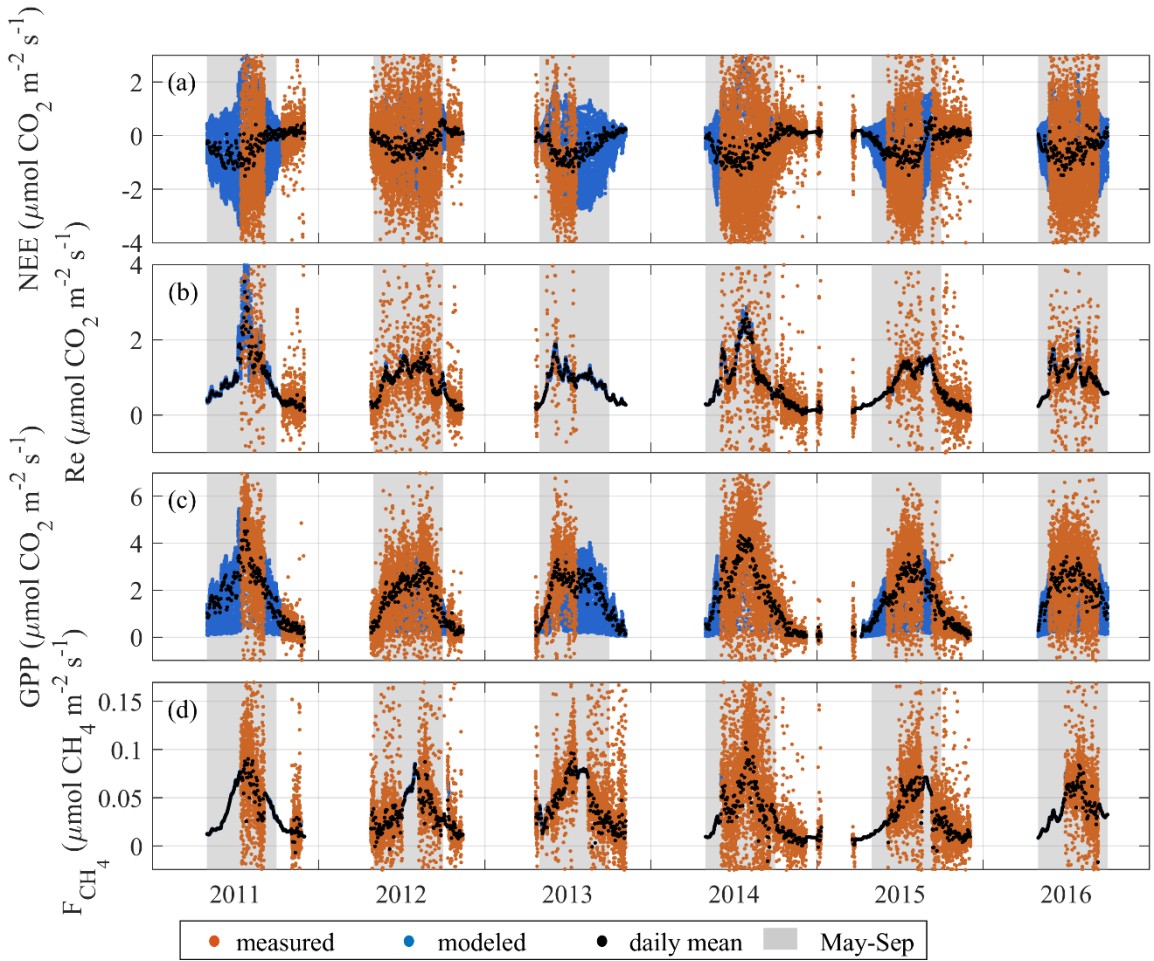

**Figure 5:** Timeseries of measured and modeled 30 min values of NEE, Re, GPP and $F_{CH4}$. The gray shading highlights the growing season (May-September). The model $CH_4$ flux 30 min values are mostly hidden under the daily mean markers.

### 3.3 Drivers of the seasonal variation in $CO_2$ and $CH_4$ fluxes

Figure 6 shows the temporal course of GPP, Re and $F_{CH4}$ model parameters (Eqs. 4, 5, 7) and reveals their well-defined seasonalities and interannual differences. The $CH_4$ reference flux has pronounced maxima in April and October-November (Fig. 6a), when the emissions may have been dominated by ebullition which is not controlled by temperature (so that high efflux coincided with low peat temperature). Between June and September, $F_{ref}$ falls on average from 0.045 to 0.035 μmol $CH_4$ $m^{-2}$ $s^{-1}$. $Re_{ref}$ (Fig. 6b) peaks in May at 1.2 μmol $CO_2$ $m^{-2}$ $s^{-1}$, stagnates for the rest of the growing season, and thereafter gradually drops to about 0.5 μmol $CO_2$ $m^{-2}$ $s^{-1}$ by December. However, this behavior is again detectable only on monthly average basis. k and $P_{max}$ (Fig. 6c,d) have broad maxima in the middle of the growing season, more pronounced in $P_{max}$ than in k, in apparent relation with the LAI seasonality.

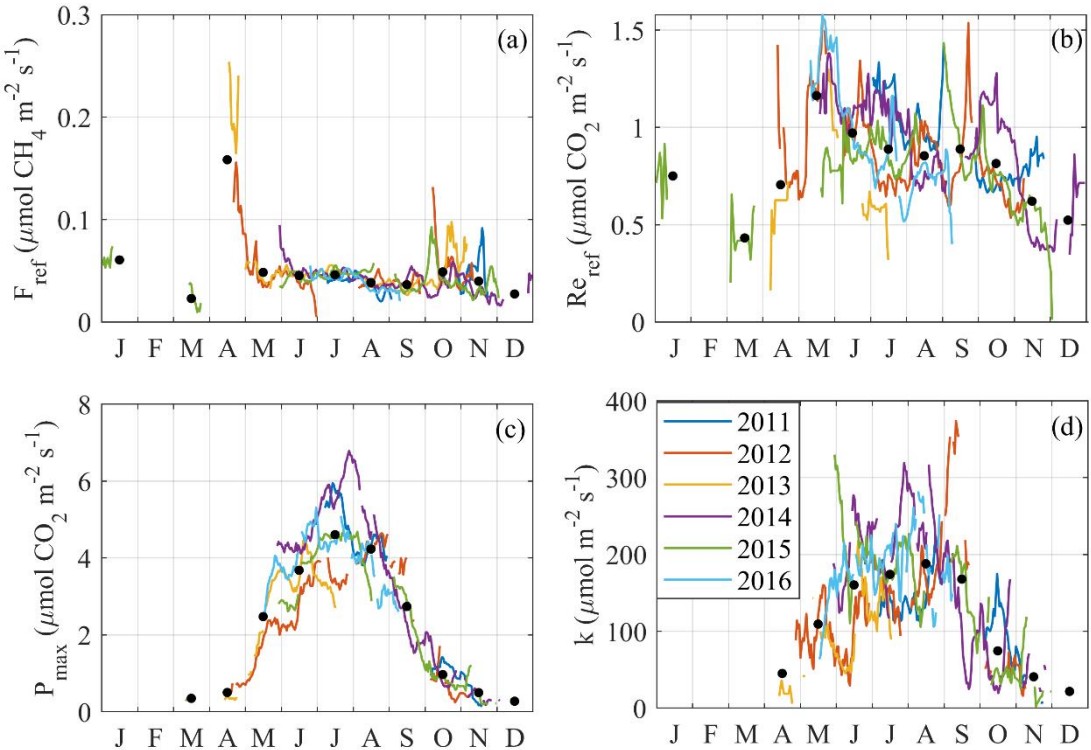

**Figure 6:** Time series of $CO_2$ and $CH_4$ flux model parameters (Eqs. 4–6): (a) $Re_{ref}$, (b) $F_{ref}$, (c) $P_{max}$, (d) k. Monthly averages across years are shown with black dots.

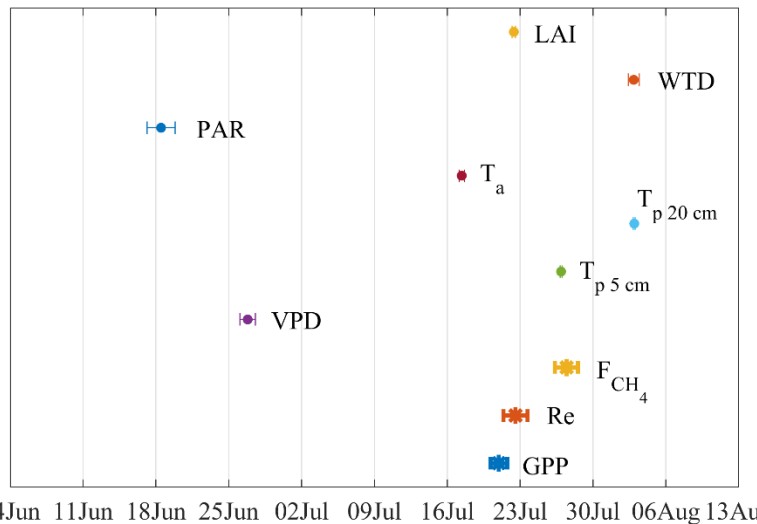

**Figure 7:** 6-year average timing of the annual peaks in fluxes and their potential drivers. The bars give the 95% CI of the peak x-value.

The timing of seasonal maxima in fluxes and environmental drivers were explored by fitting the Gaussian function to GPP, $F_{CH_4}$, Re and their physical drivers (log-linear function for $LAI_{norm}$, Wilson et al. 2007), regressed against the day
of year. The mean seasonal peak was estimated as the peak of the fit curves (Fig. 7). The peaks so derived mostly fall between the days 199 and 216, except those of VPD and PAR that occur on the days 171 and 178, respectively.

### 3.3.1 The effect of footprint variation

In heterogeneous mires, the variation in the EC footprint, controlled by wind direction and stability, may contribute to flux temporal variability, complicating the interpretation of EC fluxes (Tuovinen et al. 2019). Curiously, the two models, Kormann and Meixner (2000) and Kljun et al. (2015), produce quite similar surface compositions (Fig. 8) despite a significant difference in footprint length (Fig. 2). The variations in stability did introduce some variation in the footprint zone elongation, but did not cause significant changes in the footprint composition. The footprint composition, however,
did vary with wind direction. The pond contribution ranges directionally from zero to about 20%, being the most significant change, while the shares of the other microforms vary by about 10%.

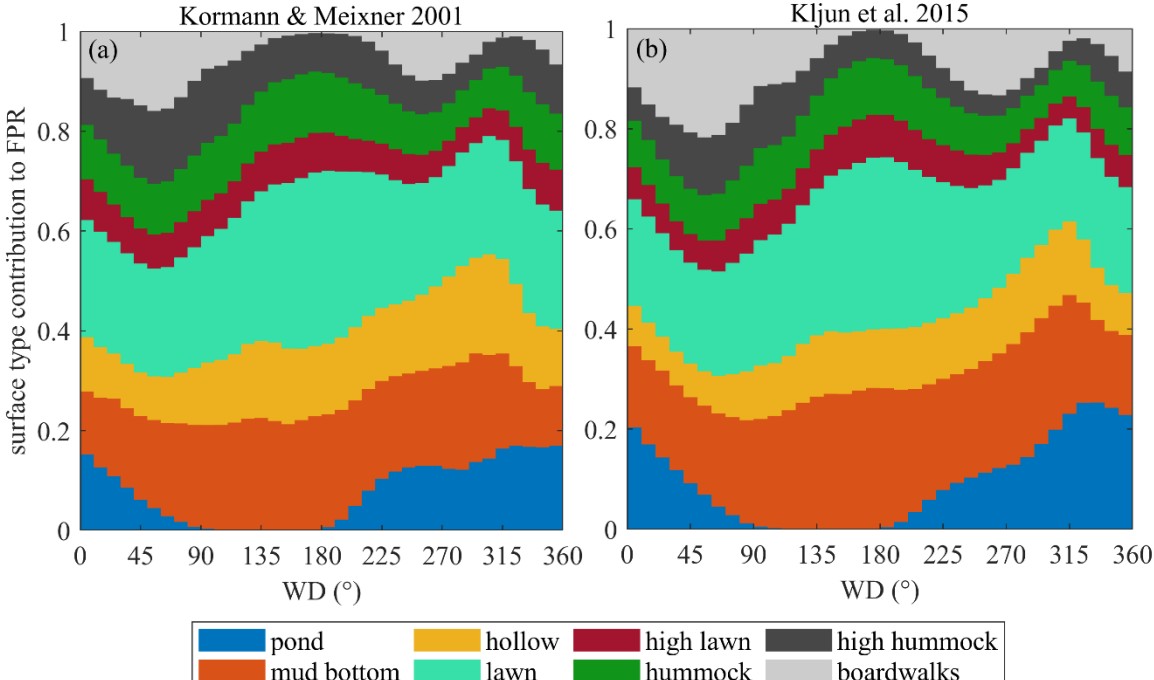

**Figure 8:** Surface type contributions to surface exchange versus wind directions, presented as means over 10˚ WD bins
using all available data.

The question whether the directional variation in footprint composition has a noticeable effect on $CO_2$ and $CH_4$ fluxes is addressed in Fig. 9 that presents both the measured fluxes and the reference flux obtained by moving-window modeling (Eqs. 4, 5, 7). The relative variation in reference fluxes was estimated as the difference between the highest and lowest
bin-averages (Fig. 9a), and proved to be broad, about 60% in $Re_{ref}$, 40% in $P_{max}$ and 20% in $F_{ref}$. The measured fluxes showed a similar directional variation (Fig. 9b). Importantly, in the western sector, roughly corresponding to the highest pond contribution, $P_{max}$ and $Re_{ref}$ show a downward peak while the $F_{ref}$ shows an upward peak.

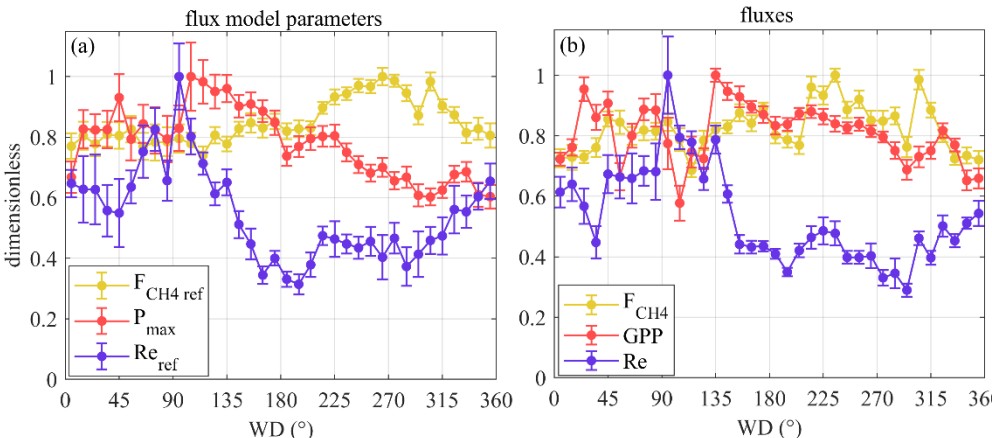

**Figure 9:** Variation of June-August fluxes and flux model parameters with wind direction. a) Variation in the $CO_2$ and $CH_4$ flux model parameters, b) measured fluxes. The markers are bin-averages normalized by the highest bin-average value for clarity.

### 3.3.2 Response to dry and hot conditions

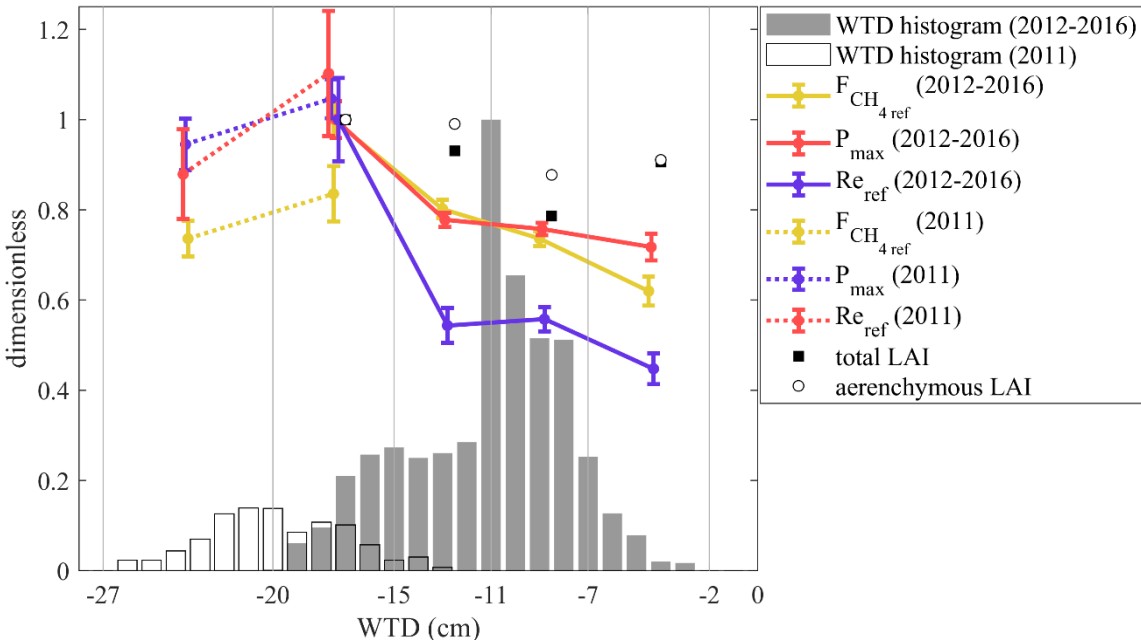

**Figure 10:** Reference fluxes for $F_{CH4}$ and Re and maximum photosynthesis Pmax *versus* water table depth, calculated as averages for the data within five WTD bins (marked by vertical lines), normalized by the value of the maximum bin-average, along with the corresponding air temperature and measured LAI bin-averages. The 2011 data are normalized by the 2012–2016 maxima, thus yielding values above unity. The data are from the period of June-August. The uncertainty bars give 50% parameter CI. The WTD histogram is shown as grey bars. WTD histograms and flux parameters are shown separately for 2011 and 2012–2016.

Our dataset includes a markedly dry period of 2011. The response of the carbon fluxes to low WTD conditions was evaluated by fitting Eqs. 4, 5, 7 in five WTD bins (Fig. 10) in order to estimate the effect of dry conditions on flux model parameters. The data of 2011 are shown separately for better contrast with the higher-WTD conditions of the other years.

When WTD is less than 15 cm, $P_{max}$, $F_{ref}$ and $Re_{ref}$ do not vary with water table. At a lower WTD (-20 > WTD > -15 cm), all of them are maximized. The two bins representing 2011 show that at yet lower WTD (<-20 cm), the reference fluxes stagnate or are slightly reduced. The parameter values in the -20…-15 cm bin, where the data of 2011 and 2012–2016 overlap, indicate similar $P_{max}$ and $Re_{ref}$, but a lower $F_{ref}$ in 2011 compared with 2012–2016.

### 3.4 Drivers of the interannual variation in $CO_2$ and $CH_4$ fluxes

The cumulative growing season (May-Sep) NEE, its components and the net $CH_4$ emission are summarized in Table 5. The typical summertime $F_{CH4}$ sums up to 4.8–6.4 g C m$^{-2}$, and in the growing season 6.4–8.4 g C m$^{-2}$. The summertime Re is 100–150 g C m$^{-2}$ and GPP about 150–200 g C m$^{-2}$, resulting in an May–Sep NEE of -20 to -64 g C m$^{-2}$. May and September make a small addition to Re and GPP, raising the average cumulative NEE from -49 C m$^{-2}$ in Jun-Aug to -61 g C m$^{-2}$ in May–Sep.

**Table 6:** Cumulative gap-filled summer and growing season $CO_2$, $CH_4$ fluxes and their 6-year averages, in g C m$^{-2}$. The relative uncertainty is calculated using the May-September 6-year mean cumulative fluxes as 10% for $F_{CH4}$, 40% for NEE, and 20% for Re and GPP. In 2013, the NEE, Re and GPP uncertainties are assumed to be double due to poor $CO_2$ EC flux data coverage. The last column gives the average cumulative fluxes with 2011 and 2013 excluded as the years with the lowest EC data coverage.

| | June-August | | | | May-September | | | |
|---|---|---|---|---|---|---|---|---|
| | $F_{CH4}$ | NEE | Re | GPP | $F_{CH4}$ | NEE | Re | GPP |
| 2011 | 5.8±0.5 | -33±24 | 151±33 | 185±46 | 7.6±0.7 | -47±24 | 197±33 | 244±46 |
| 2012 | 4.8±0.7 | -20±24 | 125±33 | 145±46 | 6.5±0.7 | -24±24 | 169±33 | 193±46 |
| 2013 | 6.4±0.7 | -64±48 | 107±66 | 171±92 | 8.4±0.7 | -72±48 | 156±66 | 228±92 |
| 2014 | 5.8±0.7 | -57±24 | 144±33 | 201±46 | 7.3±0.7 | -77±24 | 180±33 | 257±46 |
| 2015 | 4.9±0.7 | -59±24 | 103±33 | 162±46 | 6.4±0.7 | -70±24 | 146±23 | 216±46 |
| 2016 | 4.8±0.7 | -54±24 | 114±33 | 168±46 | 6.5±0.7 | -72±24 | 157±33 | 229±46 |
| 6-year average | 5.4±0.7 | -49±24 | 124±33 | 173±46 | 7.1±0.7 | -61±24 | 167±33 | 228±46 |
| average of 2012, 2014, 2015, 2016 | 5.1±0.7 | -47±24 | 122±33 | 169±46 | 6.7±0.7 | -61±24 | 163±33 | 224±46 |

We attempted to estimate the uncertainty of cumulative fluxes by comparing the results of the current gap-filling method (Sect. 2.4) with those methods that were eventually discarded. The different gap-filling approaches led to a relative variation of some tens of g C m$^{-2}$ in Re and GPP, and about 0.5 g C m$^{-2}$ in $F_{CH4}$. In the year 2013 having the worst data coverage (Table 1), the cumulative GPP estimates by different gap-filling methods ranged between 190 and 270 g C m$^{-2}$ (i.e. up to 20% relative uncertainty on the average of 235 g C m$^{-2}$), while in the rest of the years the range was much smaller (20-30 g C m$^{-2}$ around the average GPP, or about 10%). These relative uncertainties in GPP and Re, however, aggregate to a rather high relative uncertainty on May-Sep cumulative NEE, which may be estimated at 30-40% of the 6-year mean May-Sep cumulative NEE. The relative uncertainty of cumulative NEE for 2013 alone is difficult to gauge; it

may be much higher than in the other years so the net cumulative $CO_2$ uptake is quite uncertain. However, it is a likely result and is treated as such in the following. Given these considerations, the seasonal cumulative values presented in Table 6 should be taken with caution as they contain a large proportion of gap-filled data. The uncertainties presented in Table 6 are close to the 25 g C m$^{-2}$ year estimate of the error contributed by gapfilling by Moffat et al. 2007.

The year-to-year variability of the gap-filled fluxes strongly correlates with the variability of some environmental parameters (Fig. 11). This analysis uses only June-August as the period of best data coverage. High air and peat temperature favor high GPP, Re and $F_{CH4}$. At the same time, high fluxes correspond to low WTD and high VPD. This suggests that the $CO_2$ fluxes and methane emission are temperature-controlled, and the dry conditions associated with warm weather do not impose strict limitations. This is true as long as the WTD is within the tolerance limits, i.e. above about -20 cm as shown in Fig. 10 – but such dry periods did not last long enough to drastically affect the seasonal balances. In fact, the WTD limitations on fluxes are detectable in the opposite extremes – the hot and dry 2011, and the cool and moist 2012. In both years, the summer and growing season NEE was reduced when compared with the other years. Such links between WTD and fluxes are likely to exist on sub-monthly timescales, but the averaging required to reduce the random error inherent to the flux model parameters (as in Fig. 10) means that the result requires the data from prolonged periods of WTD drawdown. Therefore, the above reasoning applies mainly to the difference between the dry summer of 2011 and the other years.

In line with the previous studies (e.g. Rinne et al. 2018), the cumulative GPP and $F_{CH4}$ values are found to be positively correlated (Fig. 12a). It is more difficult to identify the links in the other flux pairs (Fig. 12b-c). We acknowledge the big contribution of model flux to these gap-filled estimates, especially in 2013 (Table 1), which may have influenced the results in Figs. 11-12.

To support the above, the same relationships were tested by plotting monthly anomalies in the fluxes versus monthly anomalies in the drivers (Appendix B). The relationships between the monthly flux and control anomalies (for the months containing <50% model data) presented in Appendix B are similar to those in Figs. 11-12. This lends further support to the evidence of strong temperature control on the fluxes and the link between GPP and $F_{CH4}$.

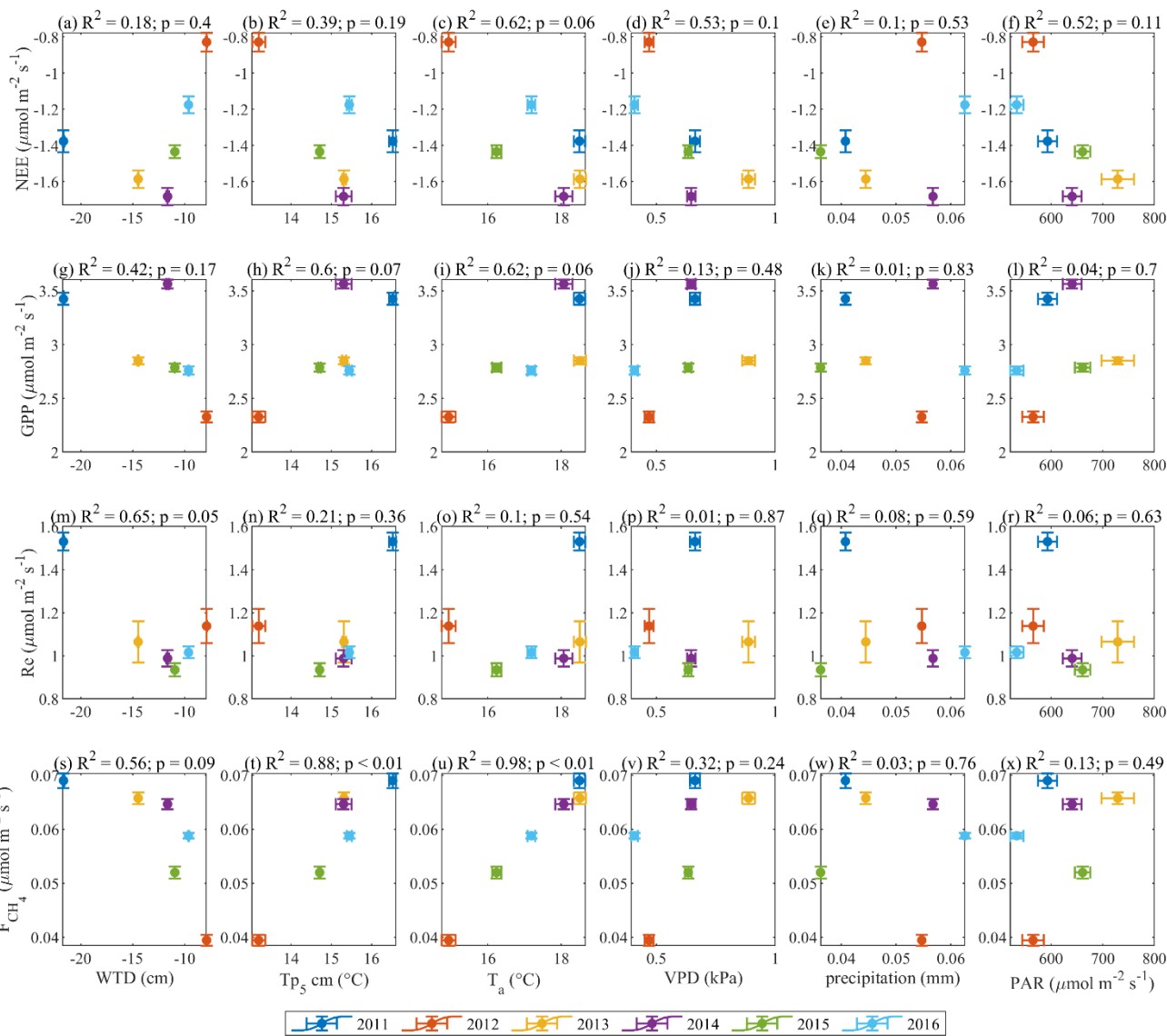

**Figure 11:** May-Sep mean gap-filled fluxes versus averages (or cumulative, in the case of precipitation) of environmental drivers.

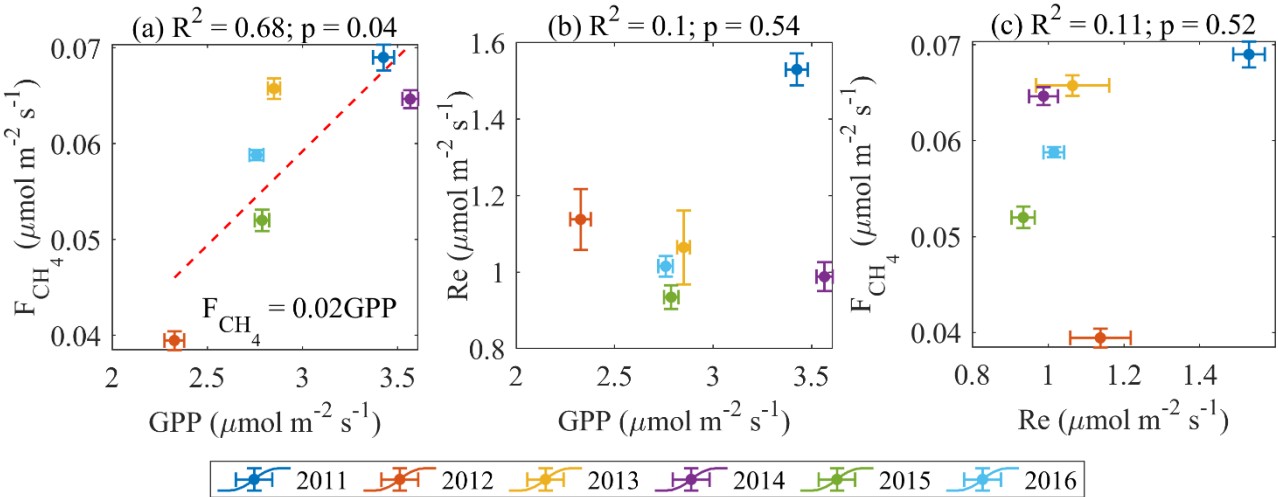

**Figure 12:** Relationships between the mean measured May-Sep fluxes.

## 4. Discussion

### 4.1 Filtering for low turbulence conditions

The friction velocity threshold was evaluated from the relationship with EC $CO_2$ and $CH_4$ fluxes (Appendix A). As this method uses all available $F_{CH4}$ data, but only nighttime $CO_2$ flux data, given equal random uncertainties one may hypothesize that using $F_{CH4}$ leads to a better-defined threshold. Here, the two thresholds, matched – as they should have, owing to identical aerodynamic transport for scalars. However, the fact that the energy balance closure deficit continues into higher $u_*$ implies the presence of some other factors degrading the performance of EC technique, which might be related to poor representativeness of the footprint, or importance of storage fluxes. This observation calls for wider application of alternative strategies to EC flux filtering, and a more critical approach to the standard $u_*$ filtering method.

### 4.2 Average growing season carbon exchange

The Siikaneva-2 bog was a net $CO_2$ sink and $CH_4$ source in each of the six growing seasons. The cumulative growing season (May–September) Re averaged $167\pm33$ g C m$^{-2}$ and GPP $228\pm46$ g C m$^{-2}$, yielding an NEE of -61±24 g C m$^{-2}$. These are within the range of the other bog studies reviewed above. For instance, middle taiga bog uptake measured near the Zotino station was between 40–60 g C m$^{-2}$ between April and October in three years (Arneth et al. 2002). Cumulative May–August NEE of 66–107 g C m$^{-2}$ was found by Humphreys et al. (2014) over two years in three Canadian bogs (Attawapiskat River, Kinoje Lake and Mer Bleu). Attawapiskat River and Kinoje Lake in Hudson Bay Lowlands are on par with Siikaneva-2, having May-August Re of about 160–170 g C m$^{-2}$, GPP of 220–240 g C m$^{-2}$ and NEE of 60–70 g C m$^{-2}$. The shrub bog Mer Bleu produced an NEE of 88–107 g C m$^{-2}$ with Re of 378–432 g C m$^{-2}$ and GPP or 466–539 g C m$^{-2}$.

The cumulative $CH_4$ efflux in Siikaneva-2 averaged 7.2 g C m$^{-2}$, which is slightly higher than the 3.6–4.6 g C m$^{-2}$ reported by Vompersky et al. 2000 based on a growing season measurement campaign (184 days in May–October) on a south taiga ridge-hollow complex in European Russia. A much higher annual release of $CH_4$ (19.5 g C m$^{-2}$) was observed by

Friborg et al. (2003) at the Plotnikovo site in the Bakchar bog (south taiga). A Canadian temperate shrub bog showed a 6-year average annual $CH_4$ emission of $3.7\pm0.5$ g C m$^{-2}$ (Roulet et al. 2007). Nadeau et al. (2013) report a cumulative summertime $CH_4$ emission equivalent to 3.9 g C m$^{-2}$ from a boreal bog in the James Bay lowlands in Canada.

The cumulative fluxes in Siikaneva-2 were close to those presented by Rinne et al. (2018) for the fen site Siikaneva-1 which is situated 1.3 km ESE of Siikaneva-2 (Fig. 1b). There, an NEE of similar magnitude was realized via Re and GPP approximately 200 g C m$^{-2}$ higher than those in Siikaneva-2 (Rinne et al. 2018). At the same time, the methane emission is about 25% lower in Siikaneva-2 bog than in Siikaneva-1 fen, estimating as the average ratio of the measured 30-min EC $CH_4$ fluxes.

### 4.3 Drivers of interannual variability

The interannual differences in cumulative GPP, Re and $CH_4$ emission were mainly controlled by temperature, with all fluxes increasing in warmer conditions, as suggested by both gap-filled growing season cumulative fluxes (Fig. 9) and monthly cumulative fluxes with the high model share months excluded (Figure B1) . In a year without positive or negative precipitation extremes, this leads to a rather stable May–Sep NEE of about -70±24 g C m$^{-2}$. In a year with ample precipitation, high WTD and low PAR (2012) this dropped to about -24±24 g C m$^{-2}$ (highly uncertain due to poor data coverage). In a year with a hot dry summer (2011), a moderately reduced May–Sep NEE of -47±24 g C m$^{-2}$ was observed. The driest summer on record (2011) neither converted the peatland into a net $CO_2$ source, nor did it arrest the $CH_4$ emission. This resilience seems to hold as long as the water table depth is above a threshold value, which is presumably at about -20 cm at this site (Fig. 9). Below that level, the reference fluxes cease increasing, implying that the positive effects of higher peat temperature have become balanced by the negative effect of dryness. However, the dry period of 2011 seems to have caused only a small decline in NEE, even though the WTD resided about 10 cm lower than the seasonal average for several weeks in a row. As warm summers clearly create conditions for high GPP, the reduction of NEE during drought is maybe more due to enhancement of respiration rather than suppression of photosynthesis. Strong enough drought can cause annual net $CO_2$ emission from bogs, however. Alm et al. (1999) report net release of carbon in a dry year at an open *Sphagnum* bog in eastern Finland, where NEE amounted to +80 g C m$^{-2}$ (GPP = 205 g C m$^{-2}$, Re = 285 g C m$^{-2}$), with most of the loss occurring on hummocks, while lawns remained nearly C-neutral. In a Swedish temperate bog, Lund et al. (2012) observed near-zero annual average NEE due to drought in two out of fours measurement years. A temperate bog became a considerable net source of $CO_2$ (Arneth et al. 2002). An apparently saturating behavior of $F_{CH_4}$ as a function of Re (Fig. 12c) is another indication of GHG emission rebalancing in dry conditions: as Re is enhanced, $F_{CH_4}$ stagnates. Our results reinforce the view that only a strong drought is able to nullify net growing season $CO_2$ sink and strongly reduce $CH_4$ emission of a boreal bog. The drought timing may be important as well – hypothetically, an early spring drought may hamper the development of deciduous (shrub, graminoid) plant biomass, and so limit photosynthesis and $CH_4$ emission for the rest of the summer; conversely, a drought initiated after the plant biomass has developed would have a lesser effect since the leaf area and the roots are already present and the vegetation has acquired a degree of resilience. This dynamics could have been in action at the Fäjemyr temperate bog, where a long and early drought lowered both GPP and Re, while a drought that started later in another year only increased Re (Lund et al. 2012)

A different set of limitations in 2012, now probably through low temperature and PAR, caused a stronger reduction in net $CO_2$ sink than in 2011. The cumulative NEE went down by about 50 g C m$^{-2}$, or about 70% of the 2013–2016 mean

NEE. A net annual emission of $F_{CH4}$ at $6.5\pm0.7$ g C m$^{-2}$ was at the low end of the range for this site, although nearly identical to the emission in 2015 and 2016. This observation suggests that lowered air temperature and overcast conditions are a stronger limiting factor to plant growth and CH$_4$ emission than moderately hot and dry conditions. Drops in solar radiation during daytime overcast/rainy weather limit photosynthesis (Nijp et al. 2015). It is difficult to tell whether the elevated WTD imposed a direct limitation on GHG production or transport. Previous studies show wet conditions without reduction in temperature result in high bog $CO_2$ uptake. Friborg et al. 2003 estimated net CO2 uptake of -108 g C m$^{-2}$ year-1 in Bakchar bog, South taiga. May-August NEE of -202 g C m$^{-2}$ was observed in a wet year with a warm spring in the Mukhrino bog in Siberian middle taiga by Alekseychik et al. 2017; dry weather in the following year resulted in significantly lower net uptake due to higher Re and lower GPP at their west Siberian site (Alekseychik et al., unpublished data). European south taiga bog Fyodorovskoye had a similar weather sensitivity, being a sink of 60 g C m$^{-2}$ in a wet year compared with a source of 25 g C m$^{-2}$ in a dry year (Arneth et al. 2002). As a side note, carbon leaching may have been higher as ample precipitation must have boosted runoff, and may have thus removed DOC which may have otherwise contributed to heterotrophic respiration and CH4 emission; Roulet et al. 2007 estimated an annual average net carbon leaching of $14.9 \pm 3.1$ g m$^{-2}$, roughly one third of the NEE observed in our study.

If the plants are expected to drive CH$_4$ emission, LAI and $F_{CH4}$ should be well correlated on a seasonal scale with closely matching peaks. This, in fact, is partly confirmed by the analysis seasonal peaks timing (Fig. 7). While LAI and GPP are in-phase (Fig. 7), the magnitudes of their peaks in the individual years were not correlated (not shown), meaning that LAI might contribute to the seasonal shape of $F_{CH4}$ but does not determine its seasonal peak. Regarding the peak in $F_{CH4}$, its occurrence between those in LAI and $T_{p20}$ is an independent indication that both controls might be important. The linear correlation between GPP and $F_{CH4}$ (Fig. 12a) comes as no surprise, as both become enhanced in warm conditions (Fig. 11), but it was not possible to check if this reflects causality of just a similar reaction to environmental factors. With the present dataset, it was not possible to confirm the plant contribution to methanogenic substrate and CH$_4$ transport. However, a pulse-labelling study of Dorodnikov et al. (2011) showed that recent photosynthates (on the time-scale of a few hours to a few days) made a minuscule contribution to the total CH$_4$ emission, whereas transport through *Eriophorum vaginatum* and *Scheuchzeria palustris* amounted to 30–50% of the total methane efflux. Consequently, the apparent relationship between $F_{CH4}$ with Net Ecosystem Productivity (NEP) might simply be due to the direct link between Leaf Area Index (LAI) and NEP. Curiously, the slope of the GPP-$F_{CH4}$ linear relationship in Fig. 13a, 0.021 (CI 0.01–0.032) is significantly lower than 0.06 reported by Rinne et al. (2018) for the June–September period in a nearby fen, which might be related to the fact that sedges, the genus usually found to enhance CH$_4$ emission, make up a larger fraction of GPP at the fen site.

## 4.4 Drivers of the seasonal $CO_2$ and CH$_4$ exchange

The carbon flux model parameters, resolved in time, display pronounced seasonal courses. The reference flux of respiration, $Re_{ref}$, peaks distinctly in May–June (Fig. 6a), which is clearly too late to indicate the release of $CO_2$ stored during the snow-on period. There is a secondary $Re_{ref}$ peak later in October. Fref is expectedly at the maximum in April–early May, at the time of snowmelt, whereas a weak correlation with LAI persists in the growing season (Fig. 6b). The autumn peaks in $F_{ref}$ are interesting, as they are not exactly correlated with those in $Re_{ref}$; transport through the sedge stems or peat freezing pushing out the bubbles are the potential mechanisms explaining the increased CH$_4$ flux (e.g. Whiting and Chanton 1992). $P_{max}$ has a well-defined bell-shaped seasonal course (Fig. 6c), with the peak coinciding with

the maximum LAI, $T_a$, $T_p$ and the lowest WTD (Fig. 7). The seasonality of k is similar to that of $P_{max}$, albeit with more scatter.

It was difficult to explain the short-term (under 1 month) variation in the four model parameters. This may have largely resulted from random uncertainty and gaps in the EC flux data, but also due to partly from the change in footprint and domination of plant species known to have different light photosynthetic light response and phenology (Korrensalo et al. 2017). The strongest positive correlation is found between $P_{max}$ and $T_{p5}$, on a biweekly time scale; this should be seen in connection to the fact that the GPP model involving $T_{p5}$ (Eq. 6) showed the best performance.

Despite the rather incomplete EC data coverage, we attempted to evaluate the importance of the non-growing season fluxes. The period of October-April was captured in a few periods, as shown in Fig 5. The magnitude of the spring peak was quite small, supplying about 4% to the summertime $CH_4$ emission in both years when it was measured. Rinne et al. (2018) found a similarly small spring peak that was absent in some years. The late autumn and winter fluxes, however, did contribute substantially (>10%) to the May-October or May-December Re and $F_{CH4}$, but less so to GPP (<5%) (Table 5).

Variations in WTD seem to cause substantial variability in the flux model parameters (Fig. 10). Note the peak in $CH_4$ and $CO_2$ reference fluxes at WTD < -15 cm in both the dry 2011 and other years. The air temperature varies significantly across the bins, but LAI stays constant. As the positive effect of Ta on GPP is known, the $P_{max}$ maximization at the warmest temperatures is expected, as long as moisture is in good supply. In contrast, the Re and $F_{CH4}$ reference fluxes are independent of temperature by virtue of their formulation. This leaves us to hypothesize that it is the photosynthesis per leaf area, expressed here via its proxy, $P_{max}$, which causes similar responses of $F_{ref}$ and $Re_{ref}$ to WTD. The fact that all three parameters become saturated only at a rather low WTD hints at why a larger reduction in GPP, respiration and methane efflux is not observed in moderately dry and hot years. Lafleur et al. (2005) observed insensitivity of Re to WTD at the Mer Bleu shrub bog. At the same time, a lowered water table position is typically shown to be a negative effect on mire $CH_4$ emission (e.g. Glagolev et al. 2001a, Kalyuzhny et al. 2009); however Rinne et al. (2018) propose the existence of an optimum WTD range (approx. -30…0 cm) maximizing the $CH_4$ efflux in a fen Siikaneva-1 (SI1) close to the bog Siikaneva-2.

## 4.5 EC Footprint variation effect

The actual EC source area is uncertain, as footprint models of Kormann & Meixner (2001) and Kljun et al. (2015) provided contrasting results regarding the footprint length (Fig. 2). Nonetheless, The EC data can be considered representative of the bog ecosystem, given the similarity between the cumulative footprint-weighted surface contributions and their abundance within the wider area (Table 2). The footprints had significantly different lengths, the Kljun et al. (2015) model yielding shorter footprints. This is fully consistent with the results of Arriga et al. (2017) who observed the same relationship, and found that the true source is between the estimates of the two models. Nevertheless, the variability in surface types between wind sectors is apparently more important than the variation with distance from the EC tower, because the surface type contributions calculated using the two models are very similar (Table 2). In such a heterogeneous site as Siikaneva bog, the directional variation in the microform contributions to EC flux does not come as a surprise (Fig. 8). The amplitude of variation in the lawn, mud bottom and pond contributions reaches 10–20%; curiously, the K&M and Kljun models agree on this, despite differing on footprint length. Maybe of greatest interest are the directional differences in the pond contribution, with the maximum in 230˚–N–30˚ and minimum at 50˚–200˚. In a similarly heterogeneous

tundra site, Tuovinen et al. (2019) found that the variation of the EC footprint with stability and wind direction induced a significant bias between the EC flux and the "true" upscaled flux of the region. Nevertheless, the cumulative contributions of the different microforms to the EC flux were close to their fractions over the 400 x 400 m area centered at the EC tower, implying that the EC data are representative of the ecosystem-average fluxes.

The question is now: is the effect of the ponds on the EC fluxes discernible, even despite their contribution varying with wind direction from 0% to a maximum of 20% depending on wind direction? In case the microforms have widely different C exchange rates, the footprint variation effect should, in principle, be discernible. Earlier chamber studies suggest that $CH_4$ efflux from microtopography elements with lower elevation (hollows) is higher than that from ridges/hummocks (e.g. Glagolev and Suvorov 2007, Glagolev and Shnyrev 2008, Maksuytov et al. 2010). Alm et al. 1999 observed annual $CH_4$ fluxes ranging from 2 g C m$^{-2}$ on hummocks to 14 g C m$^{-2}$ on hollows. High $CH_4$ emission was detected on ponds (Repo et al. 2007, Maksuytov et al. 2010) and float mats (Kazantsev et al. 2010). However, chamber studies in Siikaneva-2 bog didn't find significant differences in $CH_4$ efflux on different microforms (Korrensalo et al. 2017). Nevertheless, the EC data do show that both the normalized $F_{CH4}$ and the $CH_4$ reference flux were actually elevated within the sector with the maximum pond contribution (Fig. 9). This apparent emission peak occupies only a part of the sector containing the ponds, meaning that either the individual ponds emit different amounts of $CH_4$, or the anomaly is not related to ponds at all. The existence of the $CH_4$ emission hotspot is independently confirmed by comparison of the chamber fluxes across the chamber plots located to the west, east and north of the EC tower; the western sector $CH_4$ chamber flux is the highest (Aino Korrensalo, personal communication). The effect of ebullition may be ruled out based on the results of Männistö et al. (2019), who studied the variation of ebullitive $CH_4$ flux from vegetation-free surfaces (ponds, pond edges, mud bottoms) and estimated that the relative growing season contribution of the ebullition on these surfaces to the integrated EC-scale flux amounts to 3–5%. Boardwalks and other station infrastructure are concentrated in the W and NE sectors (Fig. 8), and appear to correlate with the $CH_4$ peak in the W sector (Fig. 9). However, the "boardwalks" share is an overestimate due rescaling the original aerial images to the map at the resolution of 1 x 1 m. Second, the disturbance of peat due to the presence of boardwalks or chamber collars was presumably minor, as the $CH_4$ emission hotspot had been present in 2011, before any infrastructure was built in the western sector, and no major traces of disturbance were seen on the mire surface. The drop in GPP and $P_{max}$ in the same sector is logical: less area covered by plants in the "pond" sector would lead to lower photosynthesis. This supports the idea of the potential pond origin of the detected $CH_4$ emission peak. Re and $Re_{ref}$ were significantly higher in the sector 0°–130° than in 150°–300°, possibly due to the higher contribution of hummocks and lower WTD in the NE sector (Fig. 9; see also Korrensalo et al. 2017).

## 5 Conclusions

The average growing season $CO_2$ and $CH_4$ cumulative fluxes in the southern Finnish bog Siikaneva-2 are within the range of the previous estimates for other boreal bogs. The 6-year average May-September NEE is -61±24 g C m$^{-2}$ which splits into 167±33 g C m$^{-2}$ of Re and 228±46 g C m$^{-2}$ of GPP, whereas the cumulative net $CH_4$ efflux is 7.2±0.7 g C m$^{-2}$. The variations in cumulative May-September Re, GPP and $F_{CH4}$ was positively correlated with the average air and peat temperatures, while water table level was not a limiting factor except in the driest of periods, which were too short to affect the seasonal balances. However, it proved difficult to separate the effects of the flux drivers due to their strong mutual dependency. The growing season cumulative GPP was well correlated with $F_{CH4}$, as in the nearby fen (Rinne et al. 2018). The non-growing season fluxes were not negligible, and need to be reassessed in future studies. However, this

mostly relates to autumn and early winter, as the early spring photosynthesis was low and the spring $F_{CH4}$ peak noticeable but its contribution was minor on the annual timescale.

Even in such a strongly patterned site, the surface heterogeneity was insufficient to demonstrate conclusively any footprint-related variations in EC fluxes. The upwind GHG sources were well integrated by the EC method, and there was not enough directional difference in surface cover. As a result, the contributions of the different surfaces did not vary with wind direction by more 10–20%, which is too small considering the joint uncertainties in the footprint model, microform map and EC fluxes. The dependency of $P_{max}$ on the ratio of vegetated/non-vegetated surfaces in footprint was the only signal of surface heterogeneity in agreement with the known EC footprint composition. EC $CH_4$ flux peaked in the western sector, possibly due to higher proportion of ponds and higher WTD in that sector, but no correlation with the surfaces contributing to the EC flux was found. The complications with the interpretation of the EC data encountered in this study stem from highly correlated changes of the environmental drivers and the flux footprint, on the timescales from week to season. These limitations apply to all eddy-covariance sites, which calls for a more critical approach to EC data interpretation and modeling, including combining EC and flux chambers, cross-validation of the EC footprint models, and the detailed analysis of the ecosystem surface composition.

**Code availability**

The codes used in the preparation of this manuscript are available upon request from the authors.

**Data availability**

Siikaneva-2 eddy-covariance and meteorological data can be downloaded from the FLUXNET repository, https://www.osti.gov/dataexplorer/biblio/dataset/1669639.

**Author contribution**

PA was partly responsible for the field measurements (eddy-covariance, meteorology and soil), post-processed and analyzed the EC data, produced the figures and wrote the manuscript text. AK conducted the vegetation sampling campaigns, analyzed the vegetation data and contributed to the text. IM helped to analyze the eddy-covariance data, and contributed to the study design and the text. SL contributed to the text and provided extensive critical commentary. EST co-managed the Siikaneva-2 site, contributed to the text and provided critical commentary. IK processed the airborne imaging and remote sensing data and contributed to the text. TV helped formulate the aims and study design, co-managed the Siikaneva-2 site, and contributed to the text.

**Competing interests**

The authors declare no competing interests.

**Acknowledgements**

PA and SL acknowledge the support of the projects CLIMOSS (Climate impacts of boreal bryophytes -from functional traits to global models) funded by the Academy of Finland, Decision no. 296116, 307192, 327180), and SOMPA (Novel soil management practices-key for sustainable bioeconomy and climate change mitigation, funded by the Strategic Research Council at the Academy of Finland, Decision no. 312912). PA is grateful for the Academy of Finland Flagship Programme for financial support to competence centre 'Forest-Human-Machine Interplay - Building Resilience,
Redefining Value Networks and Enabling Meaningful Experiences (UNITE) (decision 337655). TV, IM and IK acknowledge Academy of Finland Flagship funding (grant no. 337549). TV thanks the grant of the Tyumen region, Russia, Government in accordance with the Program of the World-Class West Siberian Interregional Scientific and Educational Center (National Project "Nauka").

**Appendix A**

Fig. A1 shows the normalized $CH_4$ and $CO_2$ fluxes and the energy balance closure plotted against $u_*$. While the normalized GHG fluxes both saturate at $u_* = 0.1$ m s$^{-1}$, the energy balance closure does so only at $u_*$ above 0.15 m s$^{-1}$. As this analysis aims to identify the variation of the turbulent heat fluxes LE and H with $u^*$, ground heat flux (G) is omitted from the
surface energy balance equation. The addition of G would lead to an increase in the SEB closure (SEBC), especially at low $u_*$ which mainly occur at night, which may conceal the $u_*$ threshold at which the SEBC starts to degrade due to the deficit in latent heat flux (LE) and sensible heat flux (H).

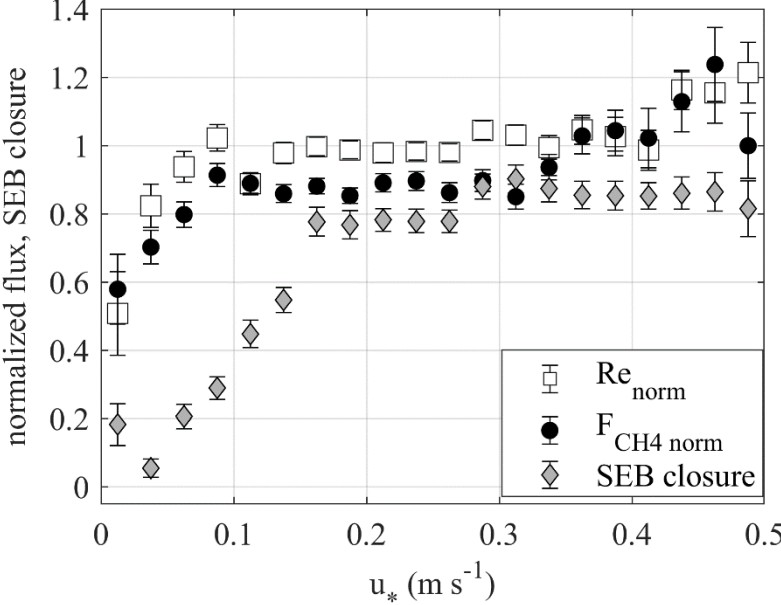

**Figure A1:** Bin-median values of $CO_2$ respiration (Re) and $CH_4$ efflux, normalized by their temperature-based statistical
models, and surface energy balance closure (SEBC = (H + LE) /Rn) *versus* friction velocity for May-October, with the bars giving STD of the median.

**Appendix B**

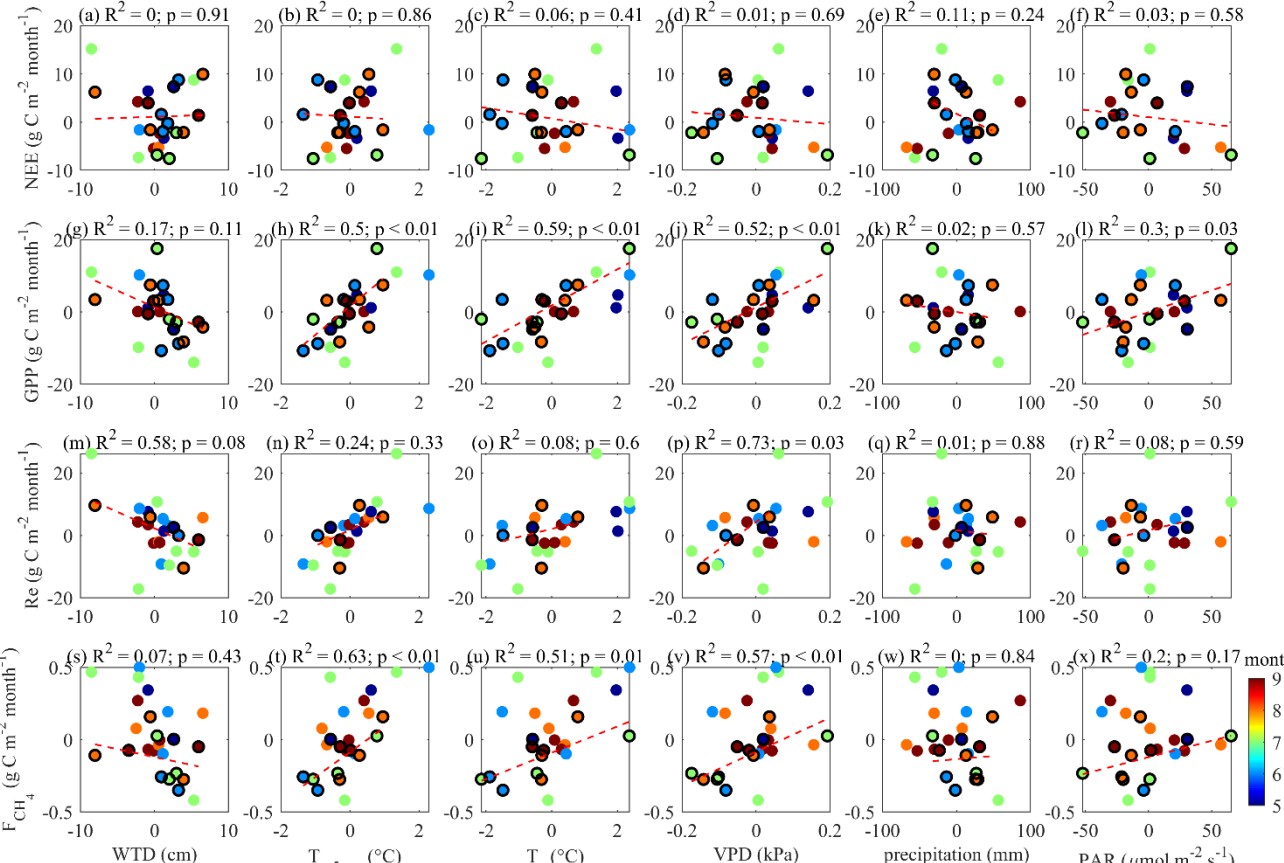

**Figure B1:** Monthly anomalies in May-September fluxes plotted versus the monthly means of the environmental drivers. The months with >50% EC flux data coverage are shown with black circles and are used for linear fitting (red dash lines). Color coding shows the month number to ensure that the seasonal cycles do not influence the presented relationships.

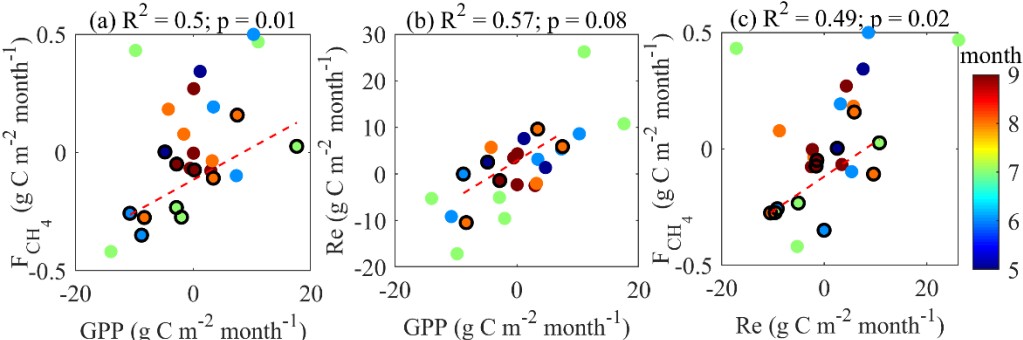

**Figure B2:** Same as Fig. B1, now for flux anomalies *versus* flux anomalies.

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
