# Peer review of "Carbon balance of a Finnish bog: temporal variability and limiting factors based on 6 years of eddy-covariance data"

_Biogeosciences, 2020_

## Author Comment (AC1)

**Response to the reviews of the preprint bg-2020-488, Alekseychik et al. 2021 "Carbon balance of a Finnish bog: temporal variability and limiting factors"**

The responses are in blue.

**Reviewer 1 (Joshua Ratcliffe)**

**General comments:**
In this study the authors present growing season (May-September) $CO_2$ and $CH_4$ flux from a boreal mire in the south of Finland. They conclude that the effect of footprint heterogeneity on fluxes is negligible and that May-September temperatures seem to be the most important factor in determining the seasonal variance of the fluxes, with warmer temperatures leading to greater net $CO_2$ uptake and greater $CH_4$ emissions.
The study is quite novel as a multi-year $CH_4$ and $CO_2$ flux record from a boreal peatland, of which only a handful exist. I like the consideration of footprint heterogeneity in the analysis, which is novel, and the degree of detail the authors have provided in the plots and the results, which make it easier than usual to assess both the quality and variability of fluxes. While the dataset contains some very large gaps, especially in 2011 and 2013, the authors have discussed this in detail and have partially considered this when drawing their conclusions from the dataset, including presenting a reasonable estimate of gapfilling error.
While the study is interesting and I would ultimately like to see it published, I have a few critical points , including one major critique about how the flux driver data has been interpreted. I am particularly concerned that the relationship between temperature and fluxes may be an artefact of the measurement gaps in the timeseries. I also think that if the temperature/flux relationship is real, then the authors should explore this in more detail, and determine whether this is related to growing season length or to more fundamental biological processes.

Dear Joshua, thank You very much for the positive evaluation of this work and the many useful comments and criticisms. We made efforts to fully address each point you have mentioned.

**Specific comments:**
My main concern is that the seasonal trend in data gaps may invalidate the analysis done in Figure 12 and thus the conclusions about the flux drivers. For example, 2013 appears 6-7 degrees warmer than 2012 in Figure 12, this must be mainly due to the differences in data coverage. Perhaps more concerning fluxes are lowest in years where the authors had the best data coverage at either end of the growing season, periods which will also have lower fluxes. As such, the same time period is not being compared in each year and naturally the fluxes are highest in the years where data is missing from the early and late season.
This is absolutely correct. These issues resulting from the use of measured fluxes should be articulated better in the discussion of Fig 12. We chose to assess the effect of the drivers on the cumulative measured fluxes, instead of on cumulative gap-filled fluxes due to the uncertainties in the model during the long gaps (this mainly concerns 2011 & 2013).
We did experiment with the gap-filled fluxes and tried using them to demonstrate the effects of environmental drivers. We abandoned method as it was difficult to tell how much role the model component played in the results, which is quite high in some of the years as you have noted. However, the versions of Fig. 12 using the original flux (currently in the draft) and using the gap-filled flux

(previous) looked similar (see below). This may also be taken as an indirect implication that our modeling approach is viable.

[Figure]

Fig. 1. Old version of Fig. 12, using gap-filled June-August fluxes.

The authors could account for this by only selecting a period where there is data in all years (July august?). Alternatively, the authors could account for the seasonal influence by looking at the anomaly for the period in question, for example presenting the value of NEE/Ta etc. for one year, minus the mean value for all other years for the months which data is available.

The suggestion to use monthly data to show the anomalies in fluxes and drivers is very good, and we will add a corresponding figure. The record is best broken down into months because of the uneven distribution of gaps.

If the temperature relationship is real I would like to see some more exploration of this. Is this effect due to growing season, in which case the authors could look at a metric such as degree days above zero, or PAR above zero, or is due to fundamental biological processes

processes? Perhaps the authors could look at light response curves of NEE during different temperature conditions in order to show this.

This is shown by the temporal trends in the NEE model parameters k and Pmax (Fig. 7) and the response of Pmax to the variation in WTD (Fig. 8).

I find it puzzling that the authors choose to talk about the u star threshold and energy balance closure in the first paragraph of the discussion. Neither of these are the focus of the study and lack of energy balance closure in peatlands is often seen and might even be expected when soil heat flux is omitted. The ustar analysis is standard. I think this can be omitted or moved to methodology section.

We would like to stress that the result shown in Fig. 2 is novel and the use of EC CH4 flux gives promise of better performance in the determination of the u∗ threshold than EC NEE. Soil heat flux is omitted as the focus is on the turbulent heat fluxes that change dynamically with u*. However, as this section appears to be out of place, we will move it to the Appendix.

The large gaps in 2011 and 2013 (appears to be around 50%) make these numbers questionable as seasonal estimates, for table 5 the authors might want to include an additional row with the averages excluding these years.

This sounds reasonable indeed, will be added.

**Technical comments:**
L41: "large area" is a bit subjective, suggest being more specific or removing – here I refer to their large proportion in the Boreal landscape land area, will be rephrased
L41: More recent estimates show that drained peatlands have now tipped this balance into a net warming effect. Suggest a qualifier such as "undrained peatlands" or "natural peatlands" – will be specified, thank You for the suggestion. Siikaneva-2 bog is a fully natural mire, and the entire discussion is certainly of relevance mainly for the natural, undrained mires.
L51-53: I agree that chambers are unsuitable for this, but it would be good to back this up a little better. Can you cite some studies that show a divergence between EC and chamber estimates? We have to admit that this statement looks somewhat EC-centric and biased. Chamber studies did produce such estimates – although the possible interpretations, of course, differ from the case of EC. The attempts to reconcile the EC with the chamber fluxes in this very site can be found in several papers by Korrensalo et al. and Männisto et al. These will be mentioned.
65-66: I agree with what is written here, i.e. "fairly wide spread" for flux totals. But in my view this contradicts several later statements L417 and L560 where the results are described as "similar to other bogs "or typical of other boreal bogs. Maybe these later statements should be amended to, "within the range seen in other boreal bogs" or "typical of **some** other boreal bogs such as x,y,z"- I totally agree, "within the range seen in other boreal bogs" sounds better, given the large spread. The basic problem here is that the "bogs" are such a diverse group of ecosystems that any averaging across them is, strictly speaking, wrong. Most studies do it anyway, due to the lack of data.
L69-71: These terms are all very subjective, warm temperature, ample sunshine etc. can they be more tightly described here? – will be rephrased. However, these are simply relative qualifiers: "favourable conditions" are such that lead to the highest net C sink possible for the given ecosystem, by definition.
L75: suggest "WTD is an important driver as it controls the thickness of the oxic zone" – will be rephrased as suggested

L98. This seems unfinished. It's been analysed in detail and what did they find? – I will make this part about the study of Tuovinen et al. (2019) more specific.

L132: "Standard schemes and quality control" is rather vague, and the cited references offer several different options in this regard, such as Moncrieff or Fratini spectral corrections. Our own work on boreal peatlands has also shown the form of timelag compensation used (optimization vs. maximisation) can have a large impact on the processed fluxes, especially when fluxes are low, and it is not clear from Sabatini et al., 2018 which of these the authors used. – The fluxes were processed using EddyUH software which is summarized in Mammarella et al. (2016) and combines the methods from other literature cited in this paragraph. Concerning the time lag approach: maximization of cross-covariance is used.

L179-180: There was presumably some impact from trampling in 2011? Can the authors state here if this was the case? I was intending to comment on this later on in the manuscript but forgot. It is an interesting question. We are only able to test for the possible effects of trampling post-factum by looking at the data. The lack in the directional different in flux model parameters between 2011 and the later years suggests a minor or absent effect of trampling; had the opposite been true, one would have observed an increase in the reference fluxes of Re and $F_{CH4}$, and maybe a drop in Pmax in the western sector starting from 2012 when the boardwalks were build. Besides, there are no significant traces of trampling around the present western boardwalks.

L217-220: I am not sure respiration or photosynthesis can be well modelled in peatlands using Q10 or Micaelis-Menthen, however depending on the site and combined with the sliding window approach it is probably acceptable. Given the gaps in the dataset I am also not convinced alternative models or techniques would perform any better. I would encourage the authors to think about using alternative techniques such as ANN or random forest in future work. – We assure you that those fluxes conform very well to the Q10- and Michaelis-Menthen-type functions and this holds throughout the season. However, we will definitely try to provide alternatives using other techniques in future studies.

L263: In one of the earlier figures the water table is shown lower than this, -25 cm. Indeed, Fig. 5e shows the lowest WTD of 2011 equal to about -25 cm. Will be corrected

L307-311: I suggest this data is presented in a table (possibly as SI) and maybe the authors can replace this with a summarised version, stating what a plausible range for the winter fluxes may be (even if this is as simple as extrapolating median, upper and lower quartile daily fluxes) – Thank You for the idea, we will organize these results as a table.

Figure 6: I really like this figure, but can the authors include monthly tick marks? I really struggled determining exactly when the gaps occurred – Sure, apologies for the difficulties with interpreting this figure, it will be improved. The ticks will be added. Note that the grey background marks the May-Sep period.

Figure 7, 8: In really like these, but I would suggest having a consistent unit of time, probably months – Fig. 8 will be remade with months in x-axis.

L371-372: it's not clear to me how summertime differs from growing season here, can you clarify this? – Summertime is June-August, growing season May-Sep. These are typically used by us but may be not as obvious to the others. To be specified.

379-390: It's great to read about the gapfilling uncertainty and how high it is, this seems entirely reasonable given the gaps in the data. Gapfiiling uncertainty is only one source of error, choice of u star threshold, filtering thresholds and measurement error are all also significant. I suggest the authors justify why only gapfilling uncertainty has been calculated and state how they think a more comprehensive assessment may differ. – Gapfilling error is maybe the easiest to estimate as it "only" requires a set of several gapfilling trials using different approaches. Moffat et al. (2007) estimate it at 25 g C / m2 year – similar to my assessment for Siikaneva-2. The other error sources are notoriously difficult to approach as they require collocated EC sensors. As shown in this prepring (Fig.1) the u* threshold is well defined so it cannot introduce

a large error. Systematic errors related to measurement and EC data postprocessing were probably smaller than that from that induced by gapfilling.

L393: please define "very low" – based on Fig. 11 I would say the fully "tolerable" WTD limit is at about -15…-20 cm.

L395: the negative impacts of what on what? Of WTD on the reference fluxes.

L295: It would be good here to talk about whether low WTD is affecting GPP or ER and by how much for how long, what is written seems really vague. Seasonal and sub-seasonal WTD variation is not in phase with the variation in fluxes, so we can only speak of seasonal averages.

L416-417: please add the calculated uncertainties to this – will be added

L415_430: This reads rather like a long list of sites and numbers with little discussion. Can the authors comment if there are any clear trends or distinctions across these sites, For instance, why is there a higher emission at Plotnikovo, or do we not know? – I will rework this section. However, the vast differences in bog subtype and vegetation cover complicate such comparisons.

L425: I am not sure the Vompersky et al., 2000 reference is appropriate, the title appears to be referencing CO2 not methane and the study also pre-dates modern Eddy Covariance measurements of CH4, Fribourg and Roulet are also rather old studies now, from the early days of Ch4 Eddy Covariance. Some more up-to-date comparisons would be good. – I will try to find more recent relevant studies, but this is basically all that is available for boreal bogs at the moment.

L433: how much is a "small decline"? – 22% of the cumulative May-Sep NEE, in relative expression (47 g C m-2 in 2011 vs. 60 g C m-2 on average). Using only Jun-Aug results in a NEE decline of 33% (understandably as the drought occurred in these months). I called this a small decline as NEE suffers from dry weather much more in some other mires.

L455: I would say your figures show this clearly, this sounds rather uncertain. – I will refer to the figures, thanks for the suggestion. I agree that this result is apparent.

L449: This sentence seems unfinished – "during drought" is missing.

L514-517: This reads like a list of different findings, can it be synthesised a little more? – I will work to make this paragraph flow better.

L557: This section seems to be missing a concluding sentence that ties it all together. – will be added.

L551: How is it that boardwalks are overestimates, compared to other features? This is not clear to me – boardwalks are thin linear features stretching over the surface of the bog. They have a width of ca. 30 cm, and so are essentially sub-grid features as the map resolution is 1 m. However, the resolution-coarsening algorithm picked them up and assigned the "boardwalks" surface type to all 1-m pixels through which they passed. This results in ca. 5-fold artificial increase in their area, which directly applies to the boardwalk contribution shown in Fig. 9. This is different from the natural surface cover, which typically consists of patches of $>1m^2$, and are also averaged over a much greater area, so the systematic error in their contribution is much smaller.

L552: but presumably, people were walking over the locations where the boardwalks were… I am not sure you can dismiss their impact for this reason - That is correct, we cannot completely rule out the possible effects of the boardwalks' installation. This will be mentioned.

L559_560: Again, if you earlier sate how variable fluxes are then say Siikaniva is typical it seems like a contradiction. Maybe re-write. – Will be rewritten.

L577: Perhaps the authors can also comment on how these limitations can be overcome? – Our attempts will be briefly summarized here, although we cannot offer a definitive solution at this point.

**Technical corrections**

L295: possible typo "from dome" – done (see the response to Reviewer 2)

L455: should be "a net annual emission" - done

L376: should be weekly to seasonally - done

---

## Author Comment (AC2)

**Response to the reviews of the preprint bg-2020-488, Alekseychik et al. 2021 "Carbon balance of a Finnish bog: temporal variability and limiting factors"**

The responses are in blue.

**Community comment 1 (Pestiaux, L., Schoenmakers, E., Thomson, L., Macfarlane, A., Griffin, S., Steel, J.)**

Dear Team of Reviewers, thank you very much for providing an assessment of our work.

**Overall summary of the paper:**
The study aimed to quantify the carbon dioxide (CO2) and methane (CH4) fluxes on boreal mires in southern Finland. It also aimed to identify the environmental factors controlling these ecosystem-atmosphere exchanges and which might be responsible for seasonal and inter-annual variability of carbon fluxes. Lastly, the study investigated if the CO2 and CH4 fluxes could help detect the heterogeneity of the surface. The study is innovative as it uses long-term data (six years of data from May to September, representing the growing seasons) measured by eddy covariance (EC) techniques.
The results of the CO2 and CH4 fluxes in the study site were similar to other boreal bogs. The variation in fluxes exchanges were driven by air and peat temperatures and the water table depth was a factor driving the atmosphere-ecosystem exchanges in dry years. Lastly, there was no relationship between CO2 and CH4 fluxes and the surface heterogeneity of the site. This was due, in part, to the uncertainty of the models used. This study will hopefully introduce further research of peat fluxes exchanges using EC techniques and will allow a better estimation and interpretations of the estimates.

**General comments:**
The different conclusions and results drawn from the study are valuable to our understanding of peat bogs dynamics. However, given the length of the paper and the amount of detail contained therein, it becomes difficult for the reader to identify the most valuable information and differentiate between this and the other findings included. We suggest that the authors could clarify the main findings they want to share with the readers and make these very apparent (e.g. a clear introductory sentence at the beginning of the section and paragraphs).
-   Thank you for noting this, I will write a paragraph with a clearer summary of the findings, and add introductory sentences.

**Dates and periods of data collection**
The paper would benefit from clarification of the exact periods from which data were collected, since the terms 'annual' and 'growing season' are used interchangeably in the paper. This can be confusing, since with the former, we would expect to see 12 months of data, and the latter, only a subset of the year. This information could be specified in:
lines 24 (what do the authors mean by "the study represents a complete series"?),
line 29 (did the authors consider data collected in winter? What do the authors mean when they say that the contribution of October-December CO2 and CH4 fluxes was 'not negligible'?),
line 62 where 'annual' is used interchangeably with 'growing season'.
Line 304, "The importance of the non-growing season fluxes was

also analyzed" meaning that annual data was indeed collected; again, reducing clarity
on exactly when the work was undertaken.

- This is a valuable comment. "Inter-annual variation" in this context means all data from a single year, covering the period of interest (June-Aug or May-Sep). Growing season is conventionally defined as May-Sep, as the time when most aboveground biomass (or LAI) develops.
- "Complete series" means a 6-year record, of course accounting for all the gaps. Will be rephrased.
- Non-negligible non-growing season fluxes: see some assessment based on fragmentary winter measurements on Lines 502-505.

**Comments on the Method section**

Line 110: We enjoyed the details to which the authors described the study sites. These detailed information enable the reader to understand better the environment in which the study was conducted.

L120: Figure 1.b could be expanded to match the size of the photo and a more detailed map of the Siikaneva-2 site with the location of the EC tower could be added. We understood that some data were gap filled with a closely situated site, Siikaneva-1. It would be valuable for the reader to have an idea of the location of Siikaneva-1 and be able to see the similarity in environmental conditions between these two sites. Are these sites similar enough to use the data interchangeably? A close-up of the map showing the replicates of the study (line 157-158) as well as the different land cover would improve the method section.

- Thank you for these suggestions. Figure 1b will be reworked and a map showing the relative locations of the Siikaneva fen and bog sites will be added. They really are very close to each other, being separated by a little over 1 km. The vegetation and peat properties are rather similar, too, of course accounting for the inevitable fen/bog differences in species composition and hydrology.

Unless the information presented in line 255, in section 3.1 (results) are information from data collected by the authors, we suggest the section (Environmental conditions) should be moved to the method section as these are background information.

- We will consider moving the Environmental conditions into the Site description section.

Some information found in the Discussion and Results sections should be explicitly set out in the method and should not be stated at the end of the paper. Line 304 (*"the importance of the non-growing season fluxes was also analyzed"),* should be stated in the Method section. – will be done

Line 254: Section 3 is called "Results and discussion". This is confusing as there is another "Discussion" section later (on Line 406). It would be clearer for the readers to have well-delimited and defined section enabling them to locate themselves in the paper.

- sorry for this inconsistency, will be corrected.

**Comments on the figures**

The authors present many figures which make it hard for the readers to understand what the most important results and main messages are. On a general note, it is easier for the reader to have the whole figure on one page and avoid the graph being cut (for example, Line 375). – Sure, will be taken care of (on the production stage if not earlier).

Line 140. Figure 2: The surface energy balance closure (SEBC) should be defined in the figure description or in-text. Whilst the formula is written (which is great), the variables

are not defined. - done

Line 145. Table 1: We do not understand why the authors
separate the periods May-September and June-August. More details on why the authors
want the readers to notice the differences would be valuable (added in the caption or in
the text where the figure is referenced). – May-Sep is the canonical "growing season" in boreal
environment studies, but we also add June-Aug as most years have good data coverage in this interval.
Some more information will be added.

Line 165. Figure 3: The FPR abbreviation could be spelled out clearly in the caption. - done

Line 285. Table 4: As said earlier, the notations in Table 4 such as (0.68...0.78) could be
clarified (at least the first time it is used in the abstract. - done

Line 314. Figure 6: We enjoyed the format of Figure 6 and the fact that the authors
highlighted some part with the part shaded in grey. The figure could be formatted slightly
bigger to allow for more precision in scales, particularly the x axis. When the figure is too
small, it is difficult to determine the variability per month. – This is understandable, done.

Line 329. Figure 8: The x axis is represented by the number of the days in the years. We
think these values are not good indicators of annual peaks. We suggest months and
dates as values in the x axis; this will make it easier for the reader
to interpret the figure. – Please note that this plot shows the timing of the peak, but the figure and/or
its discussion must be unclear, which cause this misunderstanding. To be edited.

**Comments line by line**

Line 0: We suggest the title could be more explicit. The authors could add emphasis on
the difference this study has compared to others regarding the technique used such as the
EC technique (I.e., add 'using eddy covariance technique'). We also suggest the authors
could add information about the investigation of methane balance in the title.
   -   using "EC" in the title sounds good, we will consider that.

Line 28: The authors introduce *"(6.4...8.5)"* to represent a range of data. This is done on
multiple occasion (Table 4. in Line 285). To improve the readability of the paper, a clear
explanation of what this annotation means as well as stating what the average is (I.e.,
7.1) could be added at the beginning of the paper. – this notation complies with the format of
Biogeosciences

Line 59-60: The authors specify "certain" challenges in identifying typical bogs. These
challenges could be stated clearly, and more information could be added on the reasons
the authors chose to study bogs in Siikaneva-2 site. – will be made more specific

Line 65: The author specifies that the widespread in these numbers is 'attributed to' site
specific and external factors. What are the implications of such assumptions? It would be
informative for the reader to have references for the sentence in Line 25. – The references in Lines
65-71 summarize the relevant factors, and it would be difficult to improve this part due to the scarcity
of the previous literature.

L75: The author stated that the water table level is an important driver for methane being
held in the oxic zone before it reaches that atmosphere. Explanation on why this
mechanism is important is needed. – Please note that this particular referral to the potential role of
WTD is entirely based on the literature cited in this sentence. It proved challenging to identify the
specific reaction of the $CH_4$ flux to WTD in this work, however.

L88: We noticed that the word "ebullition" was written twice in that specific sentence. – thanks a lot
for noticing that, corrected.

L133: Why are the $CH_4$ fluxes at Relative Signal Strength (RSSI) < 20 excluded from
analysis? What are the implications of this exclusion? Explanation of why this part was
excluded would benefit those less familiar with RSSI. – This is a standard quality check for open-
path analyzers. The threshold of 20 was determined based on scatter plot of $CH_4$ flux vs. RSSI.

Line 135: The sentence starting with "Interestingly...". How important is this to methods? This sentence seems not to have its place in the method, and we wonder if it should not be included in the discussion section instead? – to be moved to Discussion

Line 148: Why were these specific depths chosen for the measurement of the peat temperature? More references and/or explanations could be provided.- These depths are quite standard. -5 cm is the shallowest depth where the moss canopy can be considered more or less closed, i.e. the measurement of its T becomes possible. -50 cm approaches the greatest depth where the annual T variation can still be detected. The 20 and 35 cm depths are simply inserted between the former two.

Line 152-154: It seems that a large part of the data was taken from other sites (also Lines 162-163). We wondered to what extent the gap-filling is consistent with the other data. – the two sites from which the data were used are located nearby (Siikaneva-1 fen: 1.2 km, SMEAR-II: 7 km). Siikaneva fen has a very similar WTD dynamics and nearly identical meteorological record (not shown). SMEAR-II, too, has a meteorological record representative of the conditions at Siikaneva-2 bog.

Line 163-164: There appears to be a lot of uncertainty for the measure of LAI. It might be useful for the author to provide further information on gap-filling or discussions of these measurements. – it would be very difficult to estimate the uncertainty on LAI. For some information on this, we invite you to consult with the cited papers by Korrensalo et al.

Line 158: Further clarifications (and potentially a visual representation) of the replicated in the study should be added. We are not sure about the working out of the replicated and the total number (how can three replicates lead to 18 in total?). Do you have only one site (Siikaneva-1)? – Please see Korrensalo et al. (2017) for details on this.

Line 159: Explanation as to why LAI was measured twice a month throughout the growing season, unclear on why this number was chosen. – Please see Korrensalo et al. (2017) for details on this.

Line 180: The sentence does not read well, and we wonder if there is not a verb missing. – I do not think that a verb is missing, but I will try to rephrase it.

Line 189: The author states that the footprint lengths need 'careful calculation'. This description of the mindful calculation seems unnecessary. – sentence to be edited

Line 205: What is meant by 'high instantaneous' z0 values? Definition needed. – will be specified. Basically, this refers to 0.1 m >z0>3 m.

Line 211: We suggest that the definition or explanation of footprint nodes could be added. – This is a rather standard EC concept so I suggest not to include this.

Line 246: The author mentions a 'clearly superior performance'. It would be useful to offer a quantification, by how much? – will be specified

Line 250: The authors could clearly define what they mean by 'short gaps' and 'long gaps'. – done.

Line 366: The author mentions that "the data of 2011 is shown separately". It would be useful to provide more information in the methods section, and potentially in the results section, as to why that is the case. – To highlight the data of the drought/heatwave year 2011 and avoid their overlap with the data within the -20…-15 cm WTD bin.

Line 387: Again, "Given these considerations, the seasonal cumulative values presented in Table 5 should be taken with caution as they contain a large proportion of gap-filled data." Please explain further why gap-filled data is not an issue. – Years with a large proportion of model data are always uncertain, as model performance, especially during long gaps, is a matter of great uncertainty.

Line 410: the use of friction velocity seems to be unreliable in this case, so an explanation

of why it was used in this study would be welcome – Why, the u* threshold is well defined, in both CO2 and CH4 EC fluxes, and its application in this study is perfectly justified, as in nearly all other EC studies.

Line 412: In the sentence, "implies the presence of some other factors degrading the performance of EC technique", we wondered what other factors the authors meant. We suggest that the authors write clearly if the factors are unknown as this would make it clearer for the reader and future researchers. - Those are listed in the latter part of the same sentence.

Line 465: We notice that the word "limits" was in the sentence and seemed out of place. Is that a typing mistake? – Indeed it is, thanks for noticing that!

Line 506: The rhetorical question "what might cause such a peak in Ch4...?" may not be necessary as it could add confusion to the reader. – will be rephrased.

---

## Author Comment (AC3)

**Response to the reviews of the preprint bg-2020-488, Alekseychik et al. 2021 "Carbon balance of a Finnish bog: temporal variability and limiting factors"**

The responses are in blue.

**Reviewer 2 (anonymous)**

I was excited to read this paper given the huge amounts of data and the relatively long measurement period (6 growing seasons! Both CO2 and CH4 measurements), as well as having really all the important environmental variables and fluxes measured concurrently. I think that the conclusions are more or less supported by the results in a logical way and given what I know about the site from reading other papers (Korrensalo, et al....). However, as a peatland expert who knows a lot about C fluxes and modeling but doesn't use EC techniques, the results section manuscript was incredibly challenging to read and to follow. Fortunately, the discussion section mostly redeemed it; the authors did a nice job of integrating the results of this study with earlier studies at this site and across northern peatlands.

The challenge was that I was not convinced on the appropriateness of the modeling and the subsequent analysis of the model parameters. This is partly a result of the framing; I thought that these parameters were simply used for gap-filling (e.g. Table 5) but instead these made up the bulk of the results. An analysis of the modelling parameters used in the flux calculations comprised the whole of Sections 3.2, 3.3, 3.4, 3.5. This was problematic because the explanation of the modelling was insufficient and unclear and the justification for the approach, both theoretical and practically, was quite weak. First, the parameters are not even named (defined) or explained (Section 2.4). Then, it is not clear how these parameter values were determined. Or rather, it was clear until I read the results section and looked at Figure 7 (which shows something different than Table 4), 10, and 11 which shows that these parameters are dynamic over time, then I was completely lost. What, how and why the modeling was done in this way must be clear. This analysis comprises the bulk of the results section so the explanation needs to be clear, include the relationship with time, and can take some space.

Furthermore, I was not convinced that the model structure is completely appropriate until I read the explanation in the discussion and got a refresher about other results from Siikaneva; there is no justification for the use of these models for this site other than an earlier study used similar methods. It isn't apparent that the authors tested alternative model structures for either Re or CH4 that might include other known controls on CH4 flux (like water table). From the discussion, a bit more insight emerges as to why the authors chose these particular models for the C flux parameters but this needs to be justified in the model description section with references to the earlier studies from this site.

Thank You for making these important remarks! We regret that the model/gapfilling method description came though as overly complicated. Perhaps a clearer explanation will improve this part and provide the necessary justification.

The somewhat unconventional modeling/gapfilling method pursues two aims:
1) good performance given a dataset characterized by a mixture of long and short gaps

2) provision of temporally varying parameters which can be used to analyze the possible drivers (namely, the model parameters)

Regarding (1), the present approach is a result of a lengthy work with the Siikaneva bog dataset. We did attempt to model GPP, Re and $CH_4$ flux with a range of "single-fit" models involving WTD, $\theta$, Ta, RH, surface (~stomatal) conductance and so forth, some of which are mentioned on L292-295. However, the performance of those was questionable. There are two problems with the "single-fit" approach: a) moisture availability (WTD, $\theta$, RH), conductance ($g_s$), and other quantities of potential importance have a comples relationship with the fluxes – this was verified by residual analysis (not shown); b) the parameters of the "single-fit" models are most influenced by the seasonal cycle in the drivers, as this time scale provides the highest range in the driver values – but the drivers are highly correlated with each other on this time scale, meaning it's impossible to properly separate their effects on the fluxes (see e.g. Fig. 8).

This last leads directly to the point (2). The lack of clarity in the exact flux-driver functional relationship motivated the extensive use of the Re, GPP and $CH_4$ model parameters. We note that the reference fluxes of the exponential models of Re and CH4 emission and the maximum photosynthesis are qualitatively the same as normalized fluxes: their variation shows not the course of the actual flux, but the flux normalized by the model driver, i.e. $T_{soil}$ for $CH_4$ flux and Re, and PAR for GPP. Thus, the model parameters lack a seasonality induced by these drivers and are more informative than actual fluxes in the case of a gappy record such as ours. Rinne et al. 2007, Rinne et al. 2018 and Rinne et al. 2020 have successfully used normalized fluxes to elucidate the controls – we aimed to do something similar without copying their methodology.

Despite the apparent complexity of the modelling method explanation, it is based around a simple idea of combining two different models:
- Model 1, using the "single-fit" approach to fill the long gaps (the parameters of those are found in Table 4). As the variation of the model parameters in long gaps cannot be established, it would be the safest to assign constant values. Certainly, this leads to potential error which You have pointed out and which is acknowledged in the manuscript.
- Model 2, based on recalculation of the model parameters in a moving time window with a daily step (Eqs. 1-3). This model provides the temporally varying series of the model parameters (e.g. Fig. 7). Given the time window width of 5-15 days (differs among GPP, Re and $F_{CH4}$), the resulting parameter time series are representative of variations on a similar time scale.
- The fluxes are gapfilled using the combination of the two models.

As said above, the model parameter time series provide information similar to the mode widely used normalized fluxes. We aimed to investigate variation of the model parameters on weekly to monthly to seasonal scales and look for matching variation in the environmental parameters that caused it. It did prove to be challenging, but the insights that seemed most reasonable are discussed in the text.

We are grateful to You for suggesting several references to support the Materials & Methods and Discussion. In fact, I independently found a similar Gaussian behavior in model parameter seasonalities as shown in Rößger et al. (2019) (Scaling and balancing…). I even made a gapfilling experiment, fitting Gaussian curves to the model parameters which seemed to provide good estimates of the parameters outside the measured period or during long gaps. However, I later realized that the wide interannual variation (Fig. 7) still cannot be captured, so this Gaussian modeling doesn't have an advantage over assigning constant parameter values for the whole gap (i.e. using a single fit as detailed above).

Justification and clarification based around the arguments given above will be incorporated in the text.

Finally,
Table 1 shows that many years had only a small amount of data meeting the QC criteria. I'm a bit concerned about the circularity of using modelled fluxes (including the gap-filled data) that have been modelled given prescribed controls (with insufficient justification for the use of the models) to look at the controls of the fluxes, given that these fluxes were modelled using temperature. A more rigorous analysis is justified.

This is a reality of Siikaneva-2 and many other sites – poor data coverage caused by technical issues. The current study, to a large extent, is an exercise in interpreting a gappy dataset and trying to tease out meaningful information. Even with such a challenging dataset, one is still tempted to inquire into which drivers caused the interannual variations. An approach we took to minimize the effect of model domination was using only measured fluxes in Figs. 12-13 and the related discussion. Table 5 reports gapfilled fluxes, but it doesn't have the aspect of interannual variability.
We will add a plot of monthly averages vs. drivers (similar to Figs. 12-13), using only the month covered well enough by data.

**Specific comments:**
Introduction: I really disliked that the background for this manuscript relied only on information and background from other EC measurements in peatlands, particularly from bogs. Linking fluxes to sub-surfaces controls was originally done using chamber measurements; these have really laid the foundation for understanding environmental controls on fluxes using EC, including at this site, and some did this more than 10 years ago.

We thank you for this substantial criticism, and will improve the Introduction by adding a paragraph on the results of chamber studies and in-situ sensor campaigns.

75: Why are all these referred to as "potential" drivers? These are known drivers, at least at other sites. – Will be rephrased as appropriate.
162: Wasn't 2011 hot and dry? Is this appropriate? Using the mean LAI course is the best solution we could think of. Interannual LAI did not show any clear relationship with the environmental drivers – neither peak LAI nor mean LAI. Surprisingly, this was also true for GPP. However, the seasonality of $P_{max}$ is undoubtedly related to that of LAI.

254: This is really only the result section given that there is a later discussion section - Corrected
295: Dome? – supposed to be "from the typical dome-shape"
Section 3.3: could use figure references. How and where was this non-growing season flux determine? – Figure references will be added. The non-growing season fluxes were estimated based on the small amount of data available outside the May-September periods (can be seen in e.g. Fig. 6)
Figure 7: this is confusing (see main points above) given also Table 4. – with the explanations I have included in the beginning of my response, this should become clearer. Fig. 7 essentially shows the outcome of the second model, the one using a moving time window approach.
350-353: Confusing – these fractions refer to the range between maxima and minima of the curves plotted in Fig. 10. Rephrased.
Table 5: Why not add indicate the error here? Especially because of disclaimer on line

359? – Error estimates added

385: 30-40% of what? – this is relative uncertainty, so fraction of the seasonal cumulative NEE value.

4.1: where is this shown? – this refers to Fig. 1. As per the request of the other reviewer, we decided to make this Appendix 1. The result is novel but doesn't fit in with what follows.

433: interannual difference controlled by temperature, but how is this related to and dependent on the model used here? – as discussed above, the model does make an impact - especially in the gappy 2011 and 2013. We will additionally stress this limitation.

441: Not shown in Fig. 13 – apologies, should be Fig. 11.

481: other studies have shown differently (e.g. King et al., 1997). – This reference will be added. Admittedly, direct conflicts between the previous studies do not make things any easier!

507-8: Unclear what this paragraph is referring to? Maybe include some references to figures and tables. – I think You may have specified wrong line numbers, but if this is about the non-growing fluxes, then the relevant results were mentioned in L304-311 and Fig.6.

555: include a figure reference here. – Added, should be Fig. 10.

560: Could this be related to the gap-filling or modeling methods? – I made three attempts to do gap-filling, with different approaches, before the one described in the current preprint. All produced very similar 6-year mean May-Sep cumulative fluxes. The 30-40% relative uncertainty on NEE that was mentioned refers to the cumulative values of the individual years; the 6-year mean NEE must have a lower uncertainty which I would estimate as ±5 g C or about 10%.

Data availability: Given that this is 2021 and there are many opportunities for data publishing and indeed this is generally required, contacting the author for data is really not an acceptable route for data availability. – Of course, we understand and will publish the original data used in this MS on an official repository with a DOI.

**Additional references to consider:**

Chadburn, S. E., Aalto, T., Aurela, M., Baldocchi, D., Biasi, C., Boike, J., ... & Westermann, S. (2020). Modeled microbial dynamics explain the apparent temperature sensitivity of wetland methane emissions. Global Biogeochemical Cycles, 34(11), e2020GB006678.

Helbig, M., L. Chasmer, N. Kljun, W. Quinton, C. Treat, O. Sonnetag (2016). The positive net radiative greenhouse gas forcing of increasing methane emissions for a rapidly thawing boreal forest-wetland landscape, *Global Change Biology 23*: 2413-2427, doi: 10.1111/gcb.13520.

King, J. Y., & Reeburgh, W. S. (2002). A pulse-labeling experiment to determine the contribution of recent plant photosynthates to net methane emission in arctic wet sedge tundra. Soil Biology and Biochemistry, 34(2), 173-180.

Rößger, N., Wille, C., Veh, G., Boike, J., & Kutzbach, L. (2019). Scaling and balancing methane fluxes in a heterogeneous tundra ecosystem of the Lena River Delta. Agricultural and Forest Meteorology, 266, 243-255.

Rößger, N., Wille, C., Holl, D., Göckede, M., & Kutzbach, L. (2019). Scaling and balancing carbon dioxide fluxes in a heterogeneous tundra ecosystem of the Lena River Delta. *Biogeosciences*, *16*(13), 2591-2615.

Treat, C.C., J. Bubier, R.K. Varner, P. Crill (2007). Time-scale dependence of environmental and plant-mediated controls on CH4 flux from a temperate fen. *Journal of Geophysical Research- Biogeochemisty* 112: G01014.

---

## Author Response (AR2)

**Reviewer 1 (Joshua Ratcliffe)**

I have read the revised manuscript and the detailed response of Alekseychik et al to my earlier comments. I am happy to say that I wholeheartedly approve of the changes that have been made. Alekseychik et al., have provided extra figures in their written response and in the revised manuscript. These figures show their interpretation of the data is correct, even when accounting for the seasonal differences in data coverage across years. In my opinion the manuscript is suitable for publication without any further changes. I look forward to reading further work from Siikaneva in the coming years.

As an aside, I noticed that contrary to what is stated in the response to the reviewers the additional row with the flux means excluding the years with the worst data coverage have not been provided in table 5 (now table 6). I am happy for the authors to have the final decision on this, but I still think this addition would be beneficial and it is likely an oversight to have not included it in the revised manuscript.

Dear Joshua, the authors of this manuscript thank you for the extensive commentary you have provided and all the improvement which resulted from it. Regarding the addition to Table 5 (6), it was an oversight on my side, I simply forgot to add this extra line, but now it has been done.

Thank you,

Pavel Alekseychik et al.

**Reviewer 2 (Anonymous)**

After completing my review of the earlier manuscript and some thought about the modeling methods currently used for the temporal gap-filling that the reviewers (including me) were critical about, I think that my concerns would be best addressed using a process-based model rather than the statistical models currently used. A process-based model would allow a simpler but also more comprehensive framework for understanding these multiple years of measurements with large data gaps and would improve the rigor of the modeling approach. This would help in calculating the seasonal C balances and the results could be compared to the statistical methods currently used. Multiple years of measurements offer a chance of independent validation of the site level parameterization. Some examples of process-based models could be HIMMELI or DNDC. Particularly if these methods are used, I think it is likely that the authors will be able to revise the manuscript sufficiently to address my earlier criticisms while ultimately simplifying a complex approach. My apologies that I didn't think of this in time for my earlier review.

Dear Reviewer, I thank you for your strive to improve this manuscript and the numerous in-depth remarks you have made. I feel that the manuscript has benefitted a lot from the improved clarification of the modeling method, assessment of the uncertainty and other changes that you had requested.

Having said this, I would like to argue in favor of the present modeling approach and against changing for a process-based model. First, as we all are well aware, process-based modeling requires a substantial amount of time, effort and thinking, which is unavailable for the study at hand; it would amount for a new study of the Siikaneva-2 bog data, which will definitely see the light of day in the future. Second, process-based modeling or comparisons between different modeling approaches is outside the scope of the paper. The current paper was planned as solely an exploratory and descriptive

study of the accumulated EC data. Third, for the stated purposes, the present model seems to be quite adequate, which is now explained on LL. 297-302. The $R^2$ and model/measured flux ratios seem quite suitable for this study. Apologies for omitting these model goodness criteria in the earlier version. I have also considered the earlier experience of Raivonen et al (2017), indeed a colleague of mine, with the HIMMELI model application to the data from the Siikaneva-1 fen. They found that the model performed well in that it realistically simulated the effect of the primary drivers such as LAI, WTD, $T_a$ and $T_p$. Nevertheless, note that HIMMELI did not capture the mean level of the fluxes as well, which would be an important feature for gapfilling. I expect that the HIMMELI performance would have be similar, when run to simulate the Siikaneva-2 bog.

Given the above reasons, I would like to argue for leaving the process modeling for a dedicated future study, and retaining the present modeling/gap-filling approach in this manuscript.